# Mitigating Estimation Errors by Twin TD-Regularized Actor and Critic for Deep Reinforcement Learning

## Abstract

We address the issue of estimation bias in deep reinforcement learning (DRL) by introducing solution mechanisms that include a new, twin TD-regularized actor-critic (TDR) method. It aims at reducing both over and under estimation errors. With TDR and by combining good DRL improvements, such as distributional learning and long $N$-step surrogate stage reward (LNSS) method, we show that our new TDR-based actor-critic learning has enabled DRL methods to outperform their respective baselines in challenging environments in DeepMind Control Suite. Furthermore, they elevate TD3 and SAC respectively to a level of performance comparable to that of D4PG (the current SOTA), and they also improve the performance of D4PG to a new SOTA level measured by mean reward, convergence speed, learning success rate, and learning variance.

## 1 Introduction

Reinforcement learning (RL) has been developed for decades to provide a mathematical formalism for learning-based control. Recently, significant progress has been made to attain excellent results for a wide range of high-dimensional and continuous state-action space problems especially in robotics applications, such as robot manipulation (Andrychowicz et al., 2017), and human-robotic interaction (Liu et al., 2022; Wu et al., 2022).

However, the fundamental issue of estimation error associated with actor-critic RL (Van Hasselt et al., 2016; Duan et al., 2021) still poses great challenge. Overestimation due to, for example, using the max operator in updates has been identified and studied (Thrun & Schwartz, 1993; Duan et al., 2021). To reduce it, most efforts have focused on attaining more accurate and stable critic networks. TD3 (Fujimoto et al., 2018) applies clipped double $Q$-learning by taking the minimum between the two $Q$ estimates. SAC (Haarnoja et al., 2018) utilizes the double $Q$ network and incorporates entropy regularization in the critic objective function to ensure more exploratory behavior to help alleviate the overestimation problem. However, directly taking the minimum value of the target networks such as that in TD3 and SAC has been reported to result in an underestimation bias (Fujimoto et al., 2018).

Evaluations have revealed multiple roles of over and under estimation errors in learning. On one hand, overestimation may not always be harmful (Lan et al., 2020) as it is considered playing a role of encouraging exploration by overestimated actions. Along this line, underestimation bias may discourage exploration. If the overestimation bias occurs in a high-value region containing the optimal policy, then encouraging exploration is a good thing (Hailu & Sommer, 1999). On the other hand, overestimation bias may also cause an agent to overly explore a low-value region. This may lead to a suboptimal policy. Accordingly, an underestimation bias may discourage an agent from exploring high-value regions or avoiding low-value regions. All things considered, if estimation errors are left unchecked, they may accumulate to negatively impact policy updates as suboptimal actions may be highly rated by a suboptimal critic, reinforcing the suboptimal action in the next policy update (Fujimoto et al., 2018). Aside from the anecdotal evidence on the roles of over and under estimation, how to mitigate both of them in a principled way remains an open issue.

While several methods and evaluations have been performed and shown promising, a major tool has been mostly left out thus far. That is, it is still not clear how, and if it is possible, to further reduce

estimation errors by considering the actor given the interplay between the actor and the critic. Only a handful of approaches have been examined. As shown in (Wu et al., 2023) with demonstrated performance improvement, PAAC uses a phased actor to account for both a $Q$ value and a TD error in actor update. A double actor idea was proposed and evaluated in (Lyu et al., 2022). It takes the minimum value estimate associated with one of the two actor networks. However, directly using the minimum of the estimated values was shown resulting in an underestimation error, similar to that in TD3. Other methods, such as Entropy (Haarnoja et al., 2018; Fox et al., 2015), mutual-information (MI) (Leibfried & Grau-Moya, 2020), and Kullback-Leibler (KL) (Vieillard et al., 2020; Rudner et al., 2021) regularization, are also used to enhance policy exploration, robustness, and stability. TD-regularized actor-critic (Parisi et al., 2019) regularizes the actor only aiming to enhance the stability of the actor learning by applying a TD error (same as that in online critic updates) as a regularization term in actor updates. However, none of these methods have shown how regularization in actor may help reduce estimation error in the critic.

In this paper, we propose a new, TD-regularized (TDR) learning mechanism which includes TD-regularized double critic networks and TD-regularized actor network. This new architecture has several properties that make it ideal for the enhancements we consider. For the TD-regularized double critic network, instead of directly selecting the minimum value from twin target networks, we select the target based on the minimum TD error, which then addresses not only overestimation but underestimation problems. For the TD-regularized actor network, we formulate a new TD error to regularize actor updates to avoid a misleading critic. This regularization term helps further reduce the estimation error in critic updates. Additionally, we apply TDR combined with distributional RL (Barth-Maron et al., 2018; Bellemare et al., 2017) and LNSS reward estimation method (Zhong et al., 2022) to further improve learning stability and performance.

## 2 RELATED WORK

To shed light on the novelty of the TDR method, here we discuss double critic networks and TD error-based actor learning to provide a backdrop. We include reviews of distributional RL (Barth-Maron et al., 2018; Bellemare et al., 2017) and long-$N$-step surrogate stage (LNSS) method (Zhong et al., 2022) in Appendix A.

Double critic networks have been used in both RL (Hasselt, 2010; Zhang et al., 2017; Weng et al., 2020) and DRL (Fujimoto et al., 2018; Haarnoja et al., 2018; Van Hasselt et al., 2016). Double $Q$ learning (Hasselt, 2010; Van Hasselt et al., 2016) was the first to show reduction of overestimation bias. TD3 (Fujimoto et al., 2018) and SAC (Haarnoja et al., 2018) also were shown effective by applying clipped double $Q$-learning by using the minimum between the two $Q$ estimates. However, these methods have induced an underestimation bias problem. (Hasselt, 2010; Zhang et al., 2017; Fujimoto et al., 2018). Consequently, weighted double $Q$ learning (Zhang et al., 2017) was proposed to deal with both overestimation and underestimation biases. However, this method has not been tested in DRL context and therefore, it lacks a systematic approach to designing the weighting function.

TD error-based actor learning is expected to be effective in reducing overestimation error since it is a consistent estimate of the advantage function with lower variance, and it discriminates feedback instead of directly using $Q$ estimates. Some actor-critic variants (Crites & Barto, 1994; Bhatnagar et al., 2007) update the actor based on the sign of a TD error with a positive error preferred in policy updates. However, TD error only measures the discrepancy between the predicted value and the target value, which may not guide exploration effectively, and using TD error alone in actor update may discourage exploration and cause slow learning, especially in high-dimensional complex problems. TD-regularized actor-critic (Parisi et al., 2019) enhanced the stability of the actor update by using the same TD error (as that in online critic update) as a regularization term. However, such use of TD error may not sufficiently evaluate the critic update because it only uses the temporal difference between target and online $Q$ estimates. Additionally, the time-varying regularization coefficient was shown leading to poor convergence (Chen et al., 2017). Note also that the TD-regularized actor-critic only considered TD-regularized actor but not the critic.

**Contributions**. 1) We introduce a novel TDR mechanism that includes TD-regularized double critic networks and TD-regularized actor network. 2) Extensive experiments using DMC benchmarks show that TDR enables SOTA performance (measureed by learning speed, success rate, variance,

and converged reward) across a wide variety of control tasks, such as locomotion, classical control, and tasks with sparse rewards. 3) We also provide qualitative analysis to show that each component of TDR contributes to mitigating both over and under estimation errors.

## 3 METHOD

### 3.1 DOUBLE Q IN ACTOR-CRITIC METHOD

For a general double $Q$ actor-critic method (Fujimoto et al., 2018; Haarnoja et al., 2018). The policy $(\pi_\phi)$ is called an actor and the state-action value function $(Q_\theta(s_k, a_k))$ is called a critic where both the actor and the critic are estimated by deep neural networks with parameters $\phi$ and $\theta$, respectively.

First, consider a policy $\pi$ that is evaluated by the state-action value function below:

$$Q^\pi(s_k, a_k) = \mathbb{E}[R_k | s_k, a_k], \tag{1}$$

where $R_k = \sum_{t=k}^{\infty} \gamma^{t-k} r_t$, $s_k \sim p(\cdot \mid s_{k-1}, a_{k-1})$, $a_k = \pi_\phi(s_k)$, and $\gamma \in (0,1)$. Most actor-critic methods are based on temporal difference (TD) learning (Sutton & Barto, 2018) that updates $Q$ estimates by minimizing the TD error, which is obtained from the the difference between a target and a critic estimated value.

Next, consider typical double $Q$ methods which entail twin $Q$ networks denoted as $Q_{\theta_1}$ and $Q_{\theta_2}$. The respective twin target networks are denoted as $Q_{\theta'_1}$ and $Q_{\theta'_2}$. In the upcoming discussions, we also use $\theta$ to denote parameters in both $Q$ networks, i.e., $\theta=\{\theta_1, \theta_2\}$. The target value $y_k$ is the lesser of the two target values,

$$y_k = r_k + \gamma \min_{\zeta=1,2} Q_{\theta'_\zeta}(s_{k+1}, \pi_{\phi'}(s_{k+1})), \tag{2}$$

where by taking the minimum of the two target values, it aims to curtail overestimation of $Q$ value frequently experienced by using a single target. Thus the critic value $Q_\theta$ is updated by minimizing the loss function $(L(\theta))$ with respect to the critic weights $\theta$:

$$L(\theta) = \mathbb{E}_{s \sim p_\pi, a \sim \pi}\big[\sum_{\zeta=1,2}(y_k - Q_{\theta_\zeta}(s_k, a_k))^2\big]. \tag{3}$$

The actor weights can be updated by the deterministic policy gradient algorithm below (Silver et al., 2014), where by convention (Fujimoto et al., 2018; Haarnoja et al., 2018), $Q_{\theta_1}$ is used to update the actor weights.

$$\nabla_\phi J(\phi) = \mathbb{E}_{s \sim p_{\pi_\phi}}\left[\nabla_a Q_{\theta_1}(s_k, a_k)|_{a=\pi_\phi(s)} \nabla_\phi \pi_\phi(s)\right]. \tag{4}$$

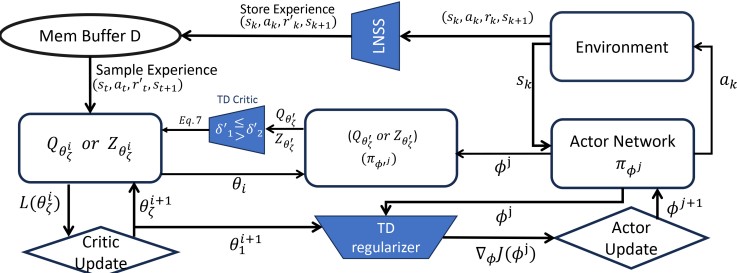

Figure 1: Twin TD-regularized Actor-Critic (TDR) Architecture

### 3.2 TWIN TD-REGULARIZED ACTOR-CRITIC (TDR) ARCHITECTURE

Figure 1 depicts our TDR-based solution mechanisms, which include twin $Q$ networks as in TD3 (Fujimoto et al., 2018) and SAC (Haarnoja et al., 2018), and an actor network. The TDR-based actor and critic updates are different from currently existing methods. In the following, we show

how the new TDR selects target value $y_k$ different from Equation (2) as used in SAC and TD3, and how that helps reduce both overestimation and underestimation errors. We also show how the new TD-regularized actor helps further reduce the estimation bias in the critic. Our TDR-based solutions in Figure 1 include two additional good improvements: distributional learning as in D4PG and long $N$-step surrogate stage (LNSS) method (Zhong et al., 2022) as described in Appendix A.

## 3.3 TD-REGULARIZED DOUBLE Q NETWORKS

To overcome overestimation, TD3 (Fujimoto et al., 2018) and SAC (Haarnoja et al., 2018) train their critic networks to minimize the loss function in Equation (3) where the target value $y_k$ is from Equation (2). While this helps reduce overestimation error, it promotes a new problem of underestimation, which usually occurs during the early stage of learning, or when subjected to corrupted reward feedback or inaccurate states.

Our TDR method aims at minimizing the same loss function as in Equation (3), but with a different target value $y_k$. Instead of directly choosing the lesser from the two target values as in Equation (2), we use the TD errors of the two target networks to set the target value. First, the two TD errors from the respective target networks are determined from:

$$\delta'_1 = r_k + \gamma Q_{\theta'_1}(s_{k+1}, \pi_{\phi'}(s_{k+1})) - Q_{\theta'_1}(s_k, a_k),$$ (5)

$$\delta'_2 = r_k + \gamma Q_{\theta'_2}(s_{k+1}, \pi_{\phi'}(s_{k+1})) - Q_{\theta'_2}(s_k, a_k).$$ (6)

The target value for TDR is then selected from the following:

$$y_k = \begin{cases} r_k + \gamma Q_{\theta'_1}(s_{k+1}, \pi_{\phi'}(s_{k+1})) & \text{if } |\delta'_1| \leq |\delta'_2|, \\ r_k + \gamma Q_{\theta'_2}(s_{k+1}, \pi_{\phi'}(s_{k+1})) & \text{if } |\delta'_1| > |\delta'_2|. \end{cases}$$ (7)

Note from Equation (7) that TDR always uses a target value associated with a smaller target TD value (regardless of the error sign) between the two. As the ultimate objective of a target network is to converge to $Q^\pi$, such choice by TDR pushes the critic via Equation (3) toward reaching the target no matter the estimation error is from above or below, but with a smaller TD value. Thus, TDR is naturally positioned to address both overesdiation and underestimation errors.

## 3.4 TD-REGULARIZED ACTOR NETWORK

Our TD-regularized actor network directly penalizes the actor's learning objective whenever there is a critic estimation error. The estimation error $\Delta^{i+1}$ of the first critic ($Q_{\theta_1}$ chosen by convention of double $Q$-based actor-critic methods) is determined from the following:

$$\Delta^{i+1} = Q_{\theta_1^{i+1}}(s_k, a_k) - (r_k + \gamma Q_{\theta_1^{i+1}}(s_{k+1}, \pi_\phi(s_{k+1}))),$$ (8)

where $i + 1$ represents the iteration number during critic update. Then the actor can be updated in the direction of maximizing $Q$ while keeping the TD error small,

$$\nabla_\phi J(\phi) = \mathbb{E}_{s \sim p_{\pi_\phi}} \left[ \nabla_a (Q_{\theta_1^{i+1}}(s_k, a_k) - \rho(\Delta^{i+1})) \Big|_{a=\pi_\phi(s)} \nabla_\phi \pi_\phi(s) \right].$$ (9)

where $\rho \in (0, 1)$ is the regularization coefficient to balance the role of TD error in the actor learning objective. Thus, we expect the TD-regularized actor to help further reduce estimation error in the critic. With TDR actor and cirtic working together hand-in-hand, TDR is positioned to help avoid bad policy updates due to a misleading $Q$ value estimate.

**Remark 1.** There are a few key differences between TDR and TD-regularized Actor Network (Parisi et al., 2019). 1) In Equation (8), they use the target critic $Q_{\theta_1^{i'}}(s_{k+1}, \pi_\phi(s_{k+1}))$ to construct TD error, the same as in critic updates. This TD error evaluates the temporal difference between target and online $Q$ estimates. To more accurately evaluate critic estimations, we construct the TD error by only using online critics which directly affects actor updates. 2) Their TD error does not sufficiently evaluate how the critic updates. Instead in Equation (8), we use the updated critic ($\theta_1^{i+1}$) to construct the TD error to directly measure critic estimation.

# 4 MITIGATING ESTIMATION BIAS BY TDR

Let $Q^\pi$ be the true $Q$ value obtained by following the current target policy $\pi$, and let $Q_\theta$ be the estimated value using neural networks. Let $\Psi_\theta^k$ be a random estimation bias. Then for state-action pairs $(s_k, a_k)$. we have,

$$Q_\theta(s_k, a_k) = Q^\pi(s_k, a_k) + \Psi_\theta^k. \tag{10}$$

The same holds for the target networks, i.e., when $\theta$ is replaced by $\theta'$ in the above equation. An overestimation problem refers to when the estimation bias $\mathbb{E}[\Psi_\theta^k] > 0$, and an underestimation problem when the estimation bias $\mathbb{E}[\Psi_\theta^k] < 0$.

## 4.1 MITIGATING ESTIMATION BIAS USING TD-REGULARIZED DOUBLE CRITIC NETWORKS

**Theorem 1.** Let $Q^\pi$ be the true $Q$ value following the current target policy $\pi$, and $Q_{\theta_1'}$ and $Q_{\theta_2'}$ be the target network estimates using double $Q$ neural networks. We assume that there exists a step random estimation bias $\psi_{\theta_\zeta'}^k$ (i.e., estimation bias at the $k$th stage), and that it is independent of $(s_k, a_k)$ with mean $\mathbb{E}[\psi_{\theta_\zeta'}^k] = \mu_\zeta', \mu_\zeta' < \infty$, for all $k$, and $\zeta = 1, 2$. Additionally, let $\delta Y_k$ denote the target value estimation error. Accordingly, we denote this error for TDR as $\delta Y_k^{TDR}$, and DQ as $\delta Y_k^{DQ}$. We then have the following,

$$|\mathbb{E}[\delta Y_k^{TDR}]| \leq |\mathbb{E}[\delta Y_k^{DQ}]|, \tag{11}$$

Where $\mathbb{E}[\delta Y_k^{TDR}] = \mathbb{E}[Q^\pi - y_k^{TDR}]$, and $\mathbb{E}[\delta Y_k^{DQ}] = \mathbb{E}[Q^\pi - y_k^{DQ}]$.

**Proof.** The proof of Theorem 1 is provided in Appendix B

**Remark 2**. By selecting a target value with less TD error, our TD-regularized double critic networks mitigate both overestimation and underestimation errors. However, vanilla double $Q$ methods usually push the target toward the lower value no matter the estimation error is over or under. Although this estimation error may not be detrimental as they may be small at each update, the presence of unchecked underestimation bias raises two concerns. Firstly, if there is no sufficient reward feedback from the environment, (e.g., for a noisy reward or sparse reward), underestimation bias may not get a chance to make corrections and may develop into a more significant bias over several updates. Secondly, this inaccurate value estimate may lead to poor policy updates in which suboptimal actions might be highly rated by the suboptimal critic, reinforcing the suboptimal action in the next policy update.

## 4.2 ADDRESSING A MISGUIDING CRITIC IN POLICY UPDATES USING TD-REGULARIZED ACTOR

**Theorem 2.** Let $Q^\pi$ denote the true $Q$ value following the current target policy $\pi$, $Q_{\theta_1}$ be the estimated value. We assume that there exists a step random estimation bias $\psi_{\theta_1}^k$ that is independent of $(s_k, a_k)$ with mean $\mathbb{E}[\psi_{\theta_1}^k] = \mu_1, \mu_1 < \infty$, for all $k$. We assume the policy is updated based on critic $Q_{\theta_1}$ using the deterministic policy gradient (DPG) as in Equation (4). Let $\delta \phi_k$ denote the change in actor parameter $\phi$ updates at stage $k$. Accordingly, we denote this change for TDR as $\delta \phi_k^{TDR}$, vanilla DPG as $\delta \phi_k^{DPG}$, and true change without any approximation error in $Q$ as $\delta \phi_k^{true}$. We then have the following,

$$\begin{cases} \mathbb{E}[\delta \phi_k^{true}] \geq \mathbb{E}[\delta \phi_k^{TDR}] \geq \mathbb{E}[\delta \phi_k^{DPG}] & \text{if } \mathbb{E}[\Psi_{\theta_1}^k] < 0, \\ \mathbb{E}[\delta \phi_k^{true}] \leq \mathbb{E}[\delta \phi_k^{TDR}] \leq \mathbb{E}[\delta \phi_k^{DPG}] & \text{if } \mathbb{E}[\Psi_{\theta_1}^k] \geq 0. \end{cases} \tag{12}$$

Where $\delta \phi_k^{true}$, $\delta \phi_k^{DPG}$, and $\delta \phi_k^{TDR}$ are defined as Equation (55),(56), and (57) respectively in Appendix B

**Proof**. The proof of Theorem 2 is provided in Appendix B.

**Remark 3**. Theorem 2, holds for $\rho \in (0, 1)$. If the regularization factor $\rho = \frac{1}{1-\gamma}$, from Equation (59), we have $\mathbb{E}[\Psi_{\theta_1}^k - \rho\Delta] = 0$ which implies that $\mathbb{E}[\delta \phi_k^{true}] = \mathbb{E}[\delta \phi_k^{TDR}]$. By using TDR, the actor will always update the same way as using the true value. While this is not realistic, the following relationship still preserves $|\mathbb{E}[\Psi_{\theta_1}^k - \rho\Delta]| \leq |\mathbb{E}[\Psi_{\theta_1}^k]|$ to help ease the negative effect of critic estimation bias.

### 4.3 Mitigating Critic Estimation Error by TD-regularized actor

**Theorem 3**. Suboptimal actor updates negatively affect the critic. Specifically, consider actor updates as in Theorem 2, in the overestimation case, we have:

$$\mathbb{E}[Q_{\theta_1}(s_k, \pi_{DPG}(s_k)] \geq \mathbb{E}[Q_{\theta_1}(s_k, \pi_{TDR}(s_k))] \geq \mathbb{E}[Q^\pi(s_k, \pi_{True}(s_k))], \qquad (13)$$

and in the underestimation case,

$$\mathbb{E}[Q_{\theta_1}(s_k, \pi_{DPG}(s_k)] \leq \mathbb{E}[Q_{\theta_1}(s_k, \pi_{TDR}(s_k))] \leq \mathbb{E}[Q^\pi(s_k, \pi_{True}(s_k))]. \qquad (14)$$

**Proof** The proof of Theorem 3 is provided in Appendix B.

**Remark 4**. For both cases, by using TD-regularized actors, it is expected to result in less estimation bias in the critic.

## 5 Experiments and Results

In this section, we provide a comprehensive evaluation of our TDR enabled actor-critic learning methods based on three commonly used, well-behaved baseline algorithms including SAC, TD3 and D4PG. Additional evaluations are also provided for popular DRL algorithms such as DDPG and PPO to provide a broader perspective on the effectiveness of TDR-based methods. All evaluations are performed based on several benchmarks in Deepmind Control Suite (Tassa et al., 2018).

In reporting evaluation results, we use the following short-form names:

**1) Base**: the original DRL algorithms including SAC, TD3, D4PG, DDPG and PPO.

**2) TDR-TD3**: Applied TD regularized double critic (TD Critic) networks, TD regularized actor (TD Actor) network, with regularization factor $\rho = 0.7$, and LNSS with $N = 100$.

**3) TDR-SAC**: Applied TD regularized double critic (TD Critic) networks, and LNSS with $N = 100$.

**4) dTDR (TDR-D4PG)**: Applied TD regularized double critic (TD Critic) network, TD regularized actor (TD Actor) network, with regularization factor $\rho = 0.7$, and LNSS with $N = 100$.

Our evaluations aim to quantitatively address the following questions:
**Q**1**.** How does TDR improve over Base and other common methods?
**Q**2**.** How does the performance of TDR methods compare to that of SOTA algorithms (D4PG)?
**Q**3**.** Is TDR method robust enough to handle both dense stochastic reward and sparse reward?
**Q**4**.** How does each component in TDR-based learning mechanisms affect performance?
**Q**5**.** How does TD regularized actor make policy updates in situations of misguiding critics?
**Q**6**.** How does the regularization coefficient $\rho$ in Equation (9) affect TD Actor performance?

Details of the implementation, training, and evaluation procedures are provided in Appendix C and D where links to all implementation codes are also provided.

### 5.1 Main Evaluation

In obtaining comprehensive evaluation results summarized in Table 1, we included a 10% noise respectively in state, action, and reward in each of the considered DMC environments in order to make the evaluations more realistic. In "Cheetah Run sparse", we sparsified the reward in the environment. All details of the environment setup can be found in Appendix C. In Table 1, "Success" is shorthand for learning success rate, "Avg. Rwd" for average reward, and "Rank" (%) is the "percent of reward difference" between the evaluated method and the SOTA D4PG, which is (the average reward of the evaluated method over that of the D4PG - 1), the more positive the better. Note that, in computing the success rate, only those trials that have achieved a reward of at least 10 are accounted for as successful learning. The results are based on the last 50 evaluations of 10 different random seeds (same for all compared algorithms). Best performances are boldfaced for average reward (Avg. Rwd). Note that we did not implement our TD Actor into SAC because SAC already has a max entropy-regulated actor.

**Q**1 **TDR improves over respective Base methods.** The learning curves for six benchmark environments are shown in Figure 2. Overall, TDR methods (solid lines) outperform their respective Base

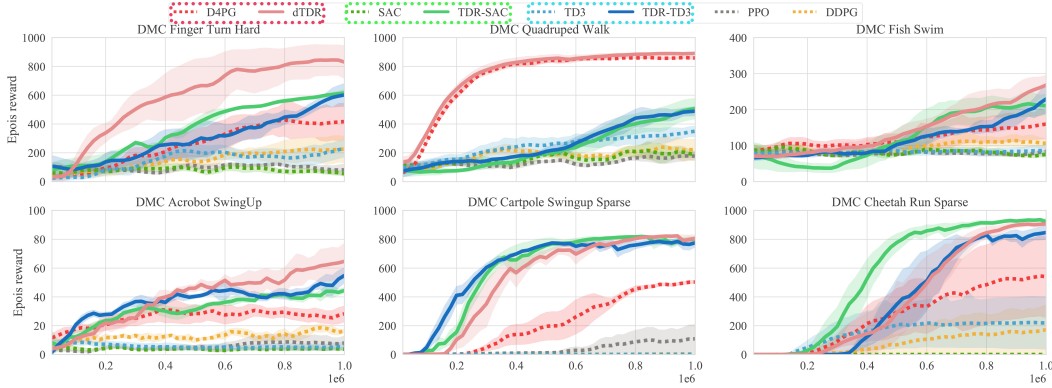

Figure 2: Systematic evaluation of TDR realized in three DRL algorithms (SAC, TD3, D4PG) in DMC environments with 10% uniform random noise in state, action, and reward. The shaded regions represent the 95 % confidence range of the evaluations over 10 seeds. The x-axis is the number of steps.

| Envirinoment | Finger Turn Hard | | | Quadruped Walk | | | Fish Swim | | |
|---|---|---|---|---|---|---|---|---|---|
| | Success [%] | Avg. Rwd $[\mu \pm 2\sigma]$ | Rank [%] | Success [%] | Avg. Rwd $[\mu \pm 2\sigma]$ | Rank [%] | Success [%] | Avg. Rwd $[\mu \pm 2\sigma]$ | Rank [%] |
| D4PG | 100 | $400.9 \pm 173.4$ | 0 | 100 | $858.5 \pm 11.4$ | 0 | 100 | $153.7 \pm 68.1$ | 0 |
| DDPG | 100 | $222.1 \pm 160.4$ | -44.6 | 100 | $226.8 \pm 133.6$ | -73.6 | 100 | $109.7 \pm 27.1$ | -28.6 |
| PPO | 100 | $85.9 \pm 50$ | -78.6 | 100 | $173.1 \pm 60.4$ | -79.8 | 100 | $78.67 \pm 6.28$ | -48.8 |
| SAC | 90 | $65.6 \pm 30.2$ | -83.6 | 100 | $196.6 \pm 73.7$ | -77.2 | 100 | $73.2 \pm 9.87$ | -52.4 |
| TD3 | 100 | $205.9 \pm 108.5$ | -48.6 | 100 | $334.8 \pm 76.4$ | -61 | 100 | $85.3 \pm 21.7$ | -44.5 |
| TDR-SAC | 100 | $601.5 \pm 147.4$ | 49.9 | 100 | $479.5 \pm 126.9$ | -44.2 | 100 | $212.3 \pm 51.2$ | 37.9 |
| TDR-TD3 | 100 | $569.8 \pm 142.1$ | 42.3 | 100 | $475.4 \pm 45.4$ | -44.6 | 100 | $204.2 \pm 41.5$ | 32.7 |
| dTDR | 100 | $\mathbf{841.02 \pm 148.3}$ | 109.8 | 100 | $\mathbf{888.6 \pm 15.7}$ | 3.46 | 100 | $\mathbf{249.9 \pm 45.5}$ | 62 |
| Envirinoment | Acrobot Swingup | | | Cartpole Swingup Sparse | | | Cheetah Run Sparse | | |
| | Success [%] | Avg. Rwd $[\mu \pm 2\sigma]$ | Rank [%] | Success [%] | Avg. Rwd $[\mu \pm 2\sigma]$ | Rank [%] | Success [%] | Avg. Rwd $[\mu \pm 2\sigma]$ | Rank [%] |
| D4PG | 100 | $26.8 \pm 8.9$ | 0 | 100 | $493.5 \pm 15.9$ | 0 | 60 | $532.8 \pm 388.4$ | 0 |
| DDPG | 100 | $17.2 \pm 3.8$ | -35.8 | 0 | $3.6 \pm 5.8$ | -99 | 50 | $160.7 \pm 284.7$ | -69.8 |
| PPO | 20 | $7.9 \pm 7.8$ | -70.5 | 80 | $99.2 \pm 172.9$ | -79.9 | 0 | $0 \pm 0$ | -100 |
| SAC | 0 | $4 \pm 2.2$ | -85.1 | 0 | $1.7 \pm 3.4$ | -99.7 | 0 | $0 \pm 0$ | -100 |
| TD3 | 0 | $5.2 \pm 4.2$ | -80.6 | 0 | $1.3 \pm 2.3$ | -99.7 | 50 | $220.5 \pm 354.7$ | -58.6 |
| TDR-SAC | 100 | $42.9 \pm 5.1$ | 60.1 | 100 | $774.2 \pm 51.1$ | 56.8 | 100 | $\mathbf{930.2 \pm 18.7}$ | 74.6 |
| TDR-TD3 | 100 | $50 \pm 7.9$ | 86.5 | 100 | $790.13 \pm 33.0$ | 60.1 | 100 | $827.8 \pm 62.2$ | 55.4 |
| dTDR | 100 | $\mathbf{62.6 \pm 14.4}$ | 133.6 | 100 | $\mathbf{810.3 \pm 34.9}$ | 64.2 | 100 | $900.1 \pm 30.8$ | 68.9 |

Table 1: Systematic evaluations of TDR respectively augmented Base algorithms. "Rank" (%) is the "percent of reward difference" between the SOTA D4PG, the more positive the better.

methods TD3, SAC and D4PG (dash lines) in terms of episode reward, learning speed, learning variance and success rate. In Table 1, among the measures, the Avg. Rwd of TDR methods outperformed respective baseline algorithms. Notice from the table that the learning success rates for all TDR methods are now 100%, a significant improvement over the Base methods. In comparison, DDPG, SAC and TD3 Base methods struggle with Acrobot Swingup, Cartpole Swingup Sparse, and Cheetah Run Sparse. Moreover, TDR methods also outperform DDPG and PPO in terms of averaged reward (Awg.Rwd), learning speed, learning variance, and success rate. Thus, TDR has helped succesfully address the random initialization challenge caused by random seeds (Henderson et al., 2018).

**Q2 TDR brings performance of Base methods close to or better than that of the SOTA D4PG.**
From Figure 2, and according to the "Rank" measure in Table 1, for all environments but Quadruped walk, TDR (TDR-SAC and TDR-TD3) helped enhance the performances of the respective Base methods. Additionally, it even outperformed the SOTA D4PG by around 40% in the "Rank" measure. For Quadruped walk, even though TDR-SAC and TDR-TD3 did not outperform D4PG, they still are the two methods, among all evaluated, that provided closest performance to D4PG. It is also

worth noting that TDR brings the performance of D4PG to a new SOTA level measured by mean reward, convergence speed, and learning success rate.

Q3 **TDR is robust under both dense stochastic reward and sparse reward.** From Figure 2 and Table 2, TDR methods outperformed their respective baselines in both dense stochastic and sparse reward in terms of average reward, learning variance, success rate, and converge speed. In particular, baseline algorithms such as TD3 and SAC struggle with sparse reward benchmark environments (cartpole swingup sparse and cheetah run sparse). However, by using TDR, they not only learned, but also achieved SOTA performance.

| | Acrobot Swingup | | Finger TurnHard | | Cartpole Swingup Sparse | |
|---|---|---|---|---|---|---|
| Methods | Avg. Rwd $[\mu \pm 2\sigma]$ | Enhancement [%] | Avg. Rwd $[\mu \pm 2\sigma]$ | Enhancement [%] | Avg. Rwd $[\mu \pm 2\sigma]$ | Enhancement [%] |
| TD3+TD Critic | $24.9 \pm 11.7$ | 378.8 | $556.2 \pm 239.8$ | 170.1 | $766.2 \pm 86.1$ | 588.4 |
| TD3+LNSS | $24.2 \pm 9.2$ | 365.4 | $547.5 \pm 120.5$ | 165.9 | $766.6 \pm 38.3$ | 588.7 |
| TD3+TD Actor | $6.9 \pm 2.9$ | 32.7 | $212.3 \pm 45.7$ | 3.1 | $339.6 \pm 231.9$ | 260.2 |
| TD3+TDR | $\mathbf{42.9 \pm 5.1}$ | 725 | $\mathbf{569.8 \pm 142.1}$ | 176.7 | $\mathbf{790.1 \pm 33.0}$ | 606.7 |
| SAC+TD Critic | $28.8 \pm 12.2$ | 620 | $588 \pm 223.8$ | 796.3 | $766.7 \pm 126.4$ | 449.6 |
| SAC+LNSS | $9.7 \pm 2.9$ | 142.5 | $573 \pm 156.5$ | 773.5 | $722.8 \pm 162.4$ | 423.7 |
| SAC+TDR | $\mathbf{42.9 \pm 5.1}$ | 972.5 | $\mathbf{601.5 \pm 147.4}$ | 816.9 | $\mathbf{774.2 \pm 51.1}$ | 454.4 |
| D4PG+TD Critic | $32.8 \pm 6.9$ | 22.4 | $835.7 \pm 140.9$ | 108.5 | $678.7 \pm 246.2$ | 37.5 |
| D4PG+LNSS | $43.9 \pm 16.7$ | 63.8 | $675.1 \pm 217.6$ | 68.4 | $759.1 \pm 31.1$ | 53.8 |
| D4PG+TD Actor | $29.9 \pm 13.8$ | 11.6 | $532.5 \pm 235.7$ | 33.5 | $600 \pm 129.3$ | 21.6 |
| dTDR | $\mathbf{62.6 \pm 14.4}$ | 133.6 | $\mathbf{841.1 \pm 148.3}$ | 109.8 | $\mathbf{810.3 \pm 34.9}$ | 64.2 |

Table 2: Systematic evaluations of each component of TDR compared to their respective Base algorithms. "Enhancement" (%) is the "percent of reward difference" between the respective Base algorithms, the larger the better. Note that TD Actor was not considered for SAC as SAC already has a max entropy-regularized actor.

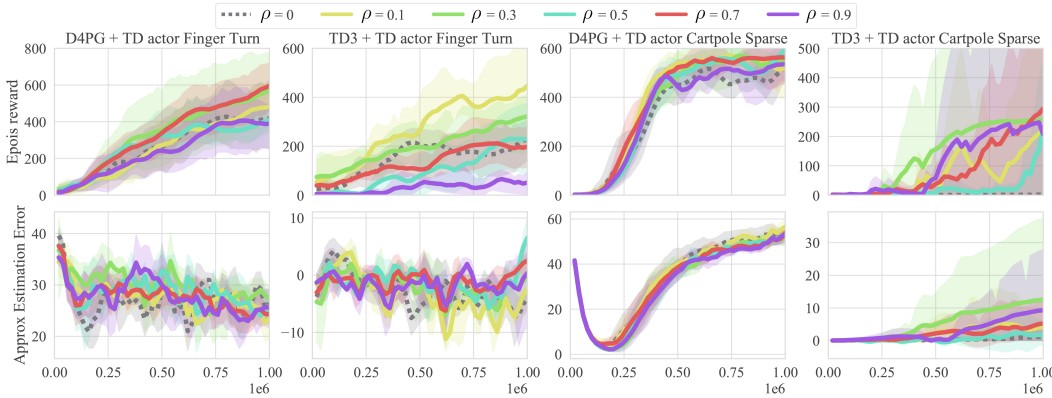

Figure 3: Evaluation of TD Actor with different $\rho$ ($\rho = 0, 0.1, 0.3, 0.5, 0.7, 0.9$) in Equations (9, 21) based on two DRL algorithms (TD3, D4PG) in DMC environments with 10% uniform random noise in state, action, and reward. The shaded regions represent the 95 % confidence range of the evaluations over 10 seeds. The x-axis is the number of steps.

## 5.2 ABLATION STUDY

To perform the ablation study, we examined TDR by removing each of the following three components. The respective short-form descriptions are:

**1) "TD Critic"**: the TD regularized double $Q$ networks.
**2) "TD Actor"**: the TD regularized actor network.
**3) "LNSS"**: LNSS method with $N = 100$.

In Table 2, "Enhancement" (%) is the "percent of reward difference" between the evaluated method and its Base method, the larger the better.

**Q4 TD Critic, TD Actor, and LNSS effectively improved the Base algorithms.** In Table 2, TD Critic, LNSS, and TD Actor all effectively improved the Base algorithms. From the table, TD Critic and LNSS have provided comparable and significant enhancement over Base algorithms. As our TD Critic methods outperform respective Base algorithms, this suggests that mitigating estimation errors both over and under from vanilla double $Q$ network is an effective way to improve performance which has also been shown in our theoretical analysis (Theorem 1). The LNSS method helped improve learning performance by reducing variances in value estimation for noisy rewards as shown both theoretically and empirically (Zhong et al., 2022). By including LNSS, our TDR is more robust under noisy and sparse rewards.

The TD Actor element also helped make appreciable improvements on learning performance as shown in Table 2. More importantly, TD Actor plays an importantly role in TDR since it not only stabilizes the policy updates as shown theoretically in Theorem 2 but also addresses the estimation error in critic as shown theoretically in Theorem 3.

### 5.3 HYPER PARAMETER STUDY

Hyperparameter study results are summarized in Figure 3 where two DRL methods (D4PG and TD3) with TD Actor are evaluated for different regularization factor $\rho$ ($\rho = 0, 0.1, 0.3, 0.5, 0.7, 0.9$). What is reported is the 10-seed averaged performance, i.e., the average of the approximate estimation error which is the difference between the true accumulated reward and the critic value: $\Psi = \frac{1}{10} \sum_{eval=0}^{9} (\sum_{t=0}^{999} \gamma^t r_t - Q(s_0, a_0))$.

**Q5 TD regularized Actor helps reduce the estimation error in critic.**

From Figure 3, with TD regularized Actor (TD Actor), the estimation errors in the critic are reduced from those without. For example, in Finger Turn hard, D4PG + TD Actor results in less overestimation error compared with $\rho = 0$ at the later stage of training. TD3 + TD Actor has less underestimation error compared with $\rho = 0$. Similarly in cartpole swingup sparse, D4PG + TD Actor results in less overestimation error compared with $\rho = 0$.

A policy can be evaluated by the "epois reward" where a higher epois reward generally results from a better policy. From Figure 3, policy updates are improved by selecting a suitable regularization factor $\rho$. Especially, in cartpole swingup sparse, TD3 + TD Actor enables successful learning whereas the Base method struggled and stuck to 0 or no learning for the entire training period.

**Q6 A range of $\rho$ ($\rho = 0.3, 0.5, 0.7$) generally are good choices.** From Figure 3, a small regularization factor $\rho = 0.1$ in TDR will result in less regularization which may not provide sufficient estimation error reduction in the critic. A larger regularization factor $\rho = 0.9$ in TDR will result in more regularization and may have a negative effect on learning. Therefore, $\rho = 0.3, 0.5, 0.7$ may be good choices. Therefore in this work, we have consistently used $\rho = 0.7$ in obtaining all results.

## 6 CONCLUSION, DISCUSSION, AND LIMITATION OF THE STUDY

1) In this work, we introduce a novel TDR mechanism that includes TD-regularized double critic networks and TD-regularized actor network. Both components are shown to help mitigate both over and under estimation errors. TDR has been shown to consistently outperform respective Base algorithms in solving benchmark tasks in terms of average reward, learning success rate, learning speed, and most times, learning variance. 2) Our analytical results also show that each component of TDR helps mitigate both over and under estimation errors. 3) As shown in Figure 2, for five out of the six environments (except quadruped walk) evaluated, our TDR combined with distributional and LNSS elements has significantly elevated the current SOTA performance of D4PG to a new level with an increase of at least 60%.

Even though we have identified a range of generally good regularization coefficient $\rho$ values $(0.3, 0.5, 0.7)$, as Figure 3 shows, different algorithms in different environments have responded somewhat differently to $\rho$. Therefore, how to effectively determine a regularization factor to have the most improvement remains a question, and thus, it is the limitation of this study. Additionally, the promising performances of TDR come after extensive training with millions of learning steps. How TDR performs under limited training time and training steps need to be further investigated.

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

## A   DISTRIBUTIONAL TDR AND LNSS

The distributional RL (Bellemare et al., 2017) represents value function in terms of probability distribution rather than function estimates. This distribution provides a more comprehensive representation of the uncertainty associated with a range of different possible reward returns and state action pairs which can provide more informative value function estimation. Many distributional RL algorithms (Bellemare et al., 2017; Dabney et al., 2018b;a) has been achieved great performance improvements on many discrete problems such as Atari benchmarks. D4PG (Barth-Maron et al., 2018) applied distributional RL into continuous control problem by combining the distributional return function within an actor-critic framework. DSAC (Duan et al., 2021) address overestimation error by applying distributional RL piggyback on SAC. Although, D4PG and DSAC can provide more accurate critic, the overestimation of actor still exists since the actor is still updated by maximizing the expectation of value function distribution. How to regulate actors in distributional RL in solving overestimations was barely discussed before.

## A.1 DISTRIBUTIONAL TD-REGULARIZED ACTOR-CRITIC (DTDR)

Here we tailor a distributional TDR (dTDR) method based on the original distributional conceptualization developed in D4PG (Barth-Maron et al., 2018; Bellemare et al., 2017). We show a number of enhancements in the meantime.

**Distributional Critic**. The distributional critic (Bellemare et al., 2017 )treated the return in Equation 1 as a random variable $Z(s_k, a_k)$ whose expectation is used as the $Q$ value estimate, namely, $Q(s_k, a_k) = \mathbb{E}[Z(s_k, a_k)]$.

In dTDR however, we use TD errors to evaluate distributional critics. Similar to Equation 5 and 6, distributional TD errors of the two target networks can be written as:

$$d'_1 = r_k + \gamma\mathbb{E}[Z_{\theta'_1}(s_{k+1}, \pi_{\phi'}(s_{k+1}))] - \mathbb{E}[Z_{\theta'_1}(s_k, a_k)], \tag{15}$$

$$d'_2 = r_k + \gamma\mathbb{E}[Z_{\theta'_2}(s_{k+1}, \pi_{\phi'}(s_{k+1}))] - \mathbb{E}[Z_{\theta'_2}(s_k, a_k)]. \tag{16}$$

The twin TD-regularized target distributional Bellman operator is thus defined as:

$$\mathcal{T}Z_k \overset{D}{=} \begin{cases} r_k + \gamma Z_{\theta'_1}(s_{k+1}, \pi_{\phi'}(s_{k+1})) & \text{if } |d'_1| \leq |d'_2| \\ r_k + \gamma Z_{\theta'_2}(s_{k+1}, \pi_{\phi'}(s_{k+1})) & \text{if } |d'_1| > |d'_2| \end{cases} \tag{17}$$

where $A \overset{D}{=} B$ denotes that two random variables $A$ and $B$ follow the same probability laws. Although the distributional Bellman operator appears similar to Equation 1, it maps state-action pairs to distributions. As such, we need to define a new TD error measure for the distribution as in D4PG (Barth-Maron et al., 2018). We consider using the following distributional loss,

$$L(\theta) = \mathbb{E}_{s \sim p_\pi, a \sim \pi}[\sum_{\zeta=1,2} l(\mathcal{T}Z_k, Z_{\theta_\zeta}(s_k, a_k))], \tag{18}$$

where $l$ measures the distance between two distributions. Many distributional RL algorithms use Kullback-Leibler (KL) divergence as the distance metric (Duan et al., 2021; Barth-Maron et al., 2018. We adopt the same metric.

**Distributional Actor**. In most distributional methods (Barth-Maron et al., 2018; Bellemare et al., 2017), policy updates are performed based on the policy gradient below,

$$\nabla_\phi J(\phi) = \mathbb{E}_{s \sim p_{\pi_\phi}}[\mathbb{E}[\nabla_a Z_\theta(s_k, a_k)]|_{a=\pi_\phi(s)}\nabla_\phi \pi_\phi(s)]. \tag{19}$$

In our dTDR, we need to use critic evaluation metrics to evaluate the quality of the current distributional critic and the regularized distributional actor. We first formulate the following loss metric:

$$L_z(\phi) = \mathbb{E}[l(r_k + \gamma Z_{\theta_1^{i+1}}(s_{k+1}, \pi_\phi(s_{k+1})), Z_{\theta_1^{i+1}}(s_k, \pi_\phi(s_k))]. \tag{20}$$

Similar to TD-regularized actor network, the distributional actor is updated in the direction of maximizing the expected critic while keeping the expected distance between the projected critic and the critic, namely,

$$\nabla_\phi J(\phi) = \mathbb{E}_{s \sim p_{\pi_\phi}}[(\mathbb{E}[\nabla_a Z_{\theta_1^{i+1}}(s_k, a_k)] - \nabla_a \rho L_z(\phi))|_{a=\pi_\phi(s)}\nabla_\phi \pi_\phi(s)], \tag{21}$$

where $\rho \in (0, 1)$ is a regularization coefficient.

## A.2 LONG N-STEP SURROGATE STAGE (LNSS) REWARD

LNSS (Zhong et al., 2022) utilizes a long reward trajectory of $N$ future steps in the estimation of stage reward $r_k$. Using the LNSS-resulted reward $r'_k$ in place of the original $r_k$ was shown to effectively reduce learning variance with significant performance improvements for off-policy methods. Given a reward trajectory of $N$ steps from time step $k$, let $G(s_{k:k+N-1}, a_{k:k+N-1}) \in \mathbf{R}$ (with shorthand notation $G_k$) denote the discounted $N$-step return, i.e.,

$$G_k = \sum_{t=k}^{k+N-1} \gamma^{t-k} r_t, \tag{22}$$

where $r_t$ is the $t$th stage reward and $t$ is from $k$ to $k + N - 1$. In LNSS, $r'_k$ is a surrogate stage reward in place of $r_k$ in Equation (2). To determine $r'_k$, LNSS treat it as a weighted average of the $N$-step reward sequence, namely

$$r'_k = \frac{\sum_{t=k}^{k+N-1} \gamma^{t-k} r_t}{\sum_{n=0}^{N-1} \gamma^n}. \tag{23}$$

As Figure 1 shows, Once $r'_k$ is obtained, it is simply used in place of $r_k$ to form a new tuple $(s_k, a_k, r'_k, s_{k+1})$, which is then stored into the memory buffer $D$. The TDR method proceeds as discussed.

## B   ESTIMATION ANALYSIS

**Lemma 1.** Let $Q^\pi$ be the true $Q$ value following the current target policy $\pi$, and $Q_{\theta'_1}$ and $Q_{\theta'_2}$ be the target network estimates using double $Q$ neural networks. We assume that there exists a step random estimation bias $\psi_{\theta'_\zeta}^k$ (i.e., estimation bias at the $k$th stage), and that it is independent of $(s_k, a_k)$ with mean $\mathbb{E}[\psi_{\theta'_\zeta}^k] = \mu'_\zeta, \mu'_\zeta < \infty$, for all $k$, and $\zeta = 1, 2$. Then for $\delta'_1$ and $\delta'_2$ respectively defined in Equations (5) and (6), we have,

$$\begin{aligned} \mathbb{E}[\delta'_1] &= -\mu'_1, \\ \mathbb{E}[\delta'_2] &= -\mu'_2. \end{aligned} \tag{24}$$

**Proof.** With the step random estimation bias $\psi_{\theta'_\zeta}^k$, We can rewrite the expectation of $\Psi_{\theta'_\zeta}^k$ as

$$\mathbb{E}[\Psi_{\theta'_\zeta}^{k+1}] = \sum_{t=k+1}^\infty \gamma^{t-k-1} \mathbb{E}[\psi_{\theta'_\zeta}^t] = \frac{1}{1-\gamma} \mu'_\zeta. \tag{25}$$

Then the expectation of the target can be written as,

$$\begin{aligned} \mathbb{E}[y_k] &= \mathbb{E}[r_k] + \gamma \mathbb{E}[(Q^\pi(s_{k+1}, a_{k+1}) + \Psi_{\theta'_\zeta}^{k+1})] \\ &= \mathbb{E}[r_k] + \gamma (\mathbb{E}[\sum_{t=k+1}^\infty \gamma^{t-k-1} r_t]) + \frac{\gamma}{1-\gamma} \mu'_\zeta \\ &= Q^\pi(s_k, a_k) + \frac{\gamma}{1-\gamma} \mu'_\zeta. \end{aligned} \tag{26}$$

By using Equations (10), and (26), the TD errors of the two target critics (Equations 5 and 6), respectably are:

$$\begin{aligned} \mathbb{E}[\delta'_1] &= \mathbb{E}[r_k] + \gamma \mathbb{E}[Q_{\theta'_1}(s_{k+1}, \pi_{\phi'}(s_{k+1}))] - \mathbb{E}[Q_{\theta'_1}(s_k, a_k)] \\ &= Q^\pi(s_k, a_k) + \frac{\gamma}{1-\gamma} \mu'_1 - Q^\pi(s_k, a_k) - \frac{1}{1-\gamma} \mu'_1 \\ &= -\mu'_1. \end{aligned} \tag{27}$$
$$\text{Similarly, } \mathbb{E}[\delta'_2] = -\mu'_2.$$

Thus Lemma 1 holds.

With Lemma 1 in place, we are now ready to analyze the estimation errors by using TDR and the double $Q$ (DQ) method as in TD3 (Fujimoto et al., 2018) and SAC (Haarnoja et al., 2018).

**Theorem 1.** Let assumptions in Lemma 1 hold, and let $\delta Y_k$ denote the target value estimation error. Accordingly, we denote this error for TDR as $\delta Y_k^{TDR}$, and DQ as $\delta Y_k^{DQ}$. We then have the following,

$$|\mathbb{E}[\delta Y_k^{TDR}]| \leq |\mathbb{E}[\delta Y_k^{DQ}]|. \tag{28}$$

**Proof.** The proof is based on enumerating a total of 8 possible scenarios of estimation errors which are determined from the relationships among the two target $Q$ values and the true $Q^\pi$ value . We provide proofs for the 4 out of 8 unique scenarios below.

First note that, $\mathbb{E}[\delta Y_k^{TDR}] = \mathbb{E}[Q^\pi - y_k^{TDR}]$, and $\mathbb{E}[\delta Y_k^{DQ}] = \mathbb{E}[Q^\pi - y_k^{DQ}]$.

**Case 1**: If the target critic values and the true value $Q^\pi$ have the following relationship:

$$\mathbb{E}[Q_{\theta_1'}] < \mathbb{E}[Q_{\theta_2'}] < Q^\pi, \tag{29}$$

i.e, $Q_{\theta_1'}$ is more underestimated as

$$|\mathbb{E}[\Psi_{\theta_1'}^k]| > |\mathbb{E}[\Psi_{\theta_2'}^k]|, \tag{30}$$

that implies

$$|\mu_1'| > |\mu_2'|. \tag{31}$$

Based on Lemma 1 and Equation (7), our TDR will use $Q_{\theta_2'}$ in the target value,

$$\mathbb{E}[y_k^{TDR}] = \mathbb{E}[r_k] + \gamma\mathbb{E}[Q_{\theta_2'}(s_{k+1}, \pi_{\phi'}(s_{k+1}))]. \tag{32}$$

However for a vanilla double $Q$ network, the target value will be $Q_{\theta_1'}$,

$$\mathbb{E}[y_k^{DQ}] = \mathbb{E}[r_k] + \gamma\mathbb{E}[Q_{\theta_1'}(s_{k+1}, \pi_{\phi'}(s_{k+1}))]. \tag{33}$$

Thus based on Equation (26), the two estimation errors of the respective target values are

$$|\mathbb{E}[\delta Y_k^{TDR}]| = |\mathbb{E}[Q^\pi - y_k^{TDR}]| = |\frac{\gamma}{1-\gamma}\mu_2'|,$$
$$|\mathbb{E}[\delta Y_k^{DQ}]| = |\mathbb{E}[Q^\pi - y_k^{DQ}]| = |\frac{\gamma}{1-\gamma}\mu_1'|. \tag{34}$$

Since $|\mu_1'| > |\mu_2'|$, we have

$$|\mathbb{E}[\delta Y_k^{TDR}]| < |\mathbb{E}[\delta Y_k^{DQ}]|. \tag{35}$$

Thus identity (28) holds.

**Case 2**: If the target critic values and the true value $Q^\pi$ have the following relationship:

$$\mathbb{E}[Q_{\theta_1'}] < Q^\pi < \mathbb{E}[Q_{\theta_2'}],$$
$$|\mathbb{E}[Q^\pi - Q_{\theta_1'}]| > |\mathbb{E}[Q^\pi - Q_{\theta_2'}]|, \tag{36}$$

then $Q_{\theta_1'}$ is expected to be underestimated and $Q_{\theta_2'}$ is overestimated. Since $|\mathbb{E}[Q^\pi - Q_{\theta_1'}]| > |\mathbb{E}[Q^\pi - Q_{\theta_2'}]|$ which implies

$$|\mathbb{E}[\Psi_{\theta_1'}^k]| > |\mathbb{E}[\Psi_{\theta_2'}^k]|, \tag{37}$$

we thus have

$$|\mu_1'| > |\mu_2'|. \tag{38}$$

Based on Lemma 1 and Equation (7), our TDR will use $Q_{\theta_2'}$ in the target value:

$$\mathbb{E}[y_k^{TDR}] = \mathbb{E}[r_k] + \gamma\mathbb{E}[Q_{\theta_2'}(s_{k+1}, \pi_{\phi'}(s_{k+1}))]. \tag{39}$$

However for a vanilla double $Q$ network, the target value will use $Q_{\theta_1'}$,

$$\mathbb{E}[y_k^{DQ}] = \mathbb{E}[r_k] + \gamma\mathbb{E}[Q_{\theta_1'}(s_{k+1}, \pi_{\phi'}(s_{k+1}))]. \tag{40}$$

Based on Equation (26), the two estimation errors of the respective target values are:

$$|\mathbb{E}[\delta Y_k^{TDR}]| = |\mathbb{E}[Q^\pi - y_k^{TDR}]| = |\frac{\gamma}{1-\gamma}\mu_2'|,$$
$$|\mathbb{E}[\delta Y_k^{DQ}]| = |\mathbb{E}[Q^\pi - y_k^{DQ}]| = |\frac{\gamma}{1-\gamma}\mu_1'|, \tag{41}$$

Since $|\mu_1'| > |\mu_2'|$, we have

$$|\mathbb{E}[\delta Y_k^{TDR}]| < |\mathbb{E}[\delta Y_k^{DQ}]|. \tag{42}$$

Thus identity (28) holds.

**Case 3**: If the target critic values and the true value $Q^\pi$ has the following relationship:

$$\mathbb{E}[Q_{\theta_1'}] < Q^\pi < \mathbb{E}[Q_{\theta_2'}],$$
$$|\mathbb{E}[Q^\pi - Q_{\theta_1'}]| < |\mathbb{E}[Q^\pi - Q_{\theta_2'}]|, \tag{43}$$

then $Q_{\theta'_1}$ is expected to be underestimated and $Q_{\theta'_2}$ is overestimated. Since $|\mathbb{E}[Q^\pi - Q_{\theta'_1}]| < |\mathbb{E}[Q^\pi - Q_{\theta'_2}]|$, it implies

$$|\mathbb{E}[\Psi^k_{\theta'_1}]| < |\mathbb{E}[\Psi^k_{\theta'_2}]|, \tag{44}$$

thus we have

$$|\mu'_1| < |\mu'_2|. \tag{45}$$

Based on Lemma 1 and Equation (7), both vanilla double $Q$ network and our TDR will pick $Q_{\theta'_1}$ in the target value:

$$\mathbb{E}[y^{TDR}_k] = \mathbb{E}[y^{DQ}_k] = \mathbb{E}[r_k] + \gamma\mathbb{E}[Q_{\theta'_1}(s_{k+1}, \pi_{\phi'}(s_{k+1}))]. \tag{46}$$

Then based on Equation (26), the two estimation errors of the respective target values are:

$$|\mathbb{E}[\delta Y^{TDR}_k]| = |\mathbb{E}[\delta Y^{DQ}_k]| = |\mathbb{E}[Q^\pi - y^{TDR}_k]| = |\frac{\gamma}{1-\gamma}\mu'_1|. \tag{47}$$

We thus have

$$|\mathbb{E}[\delta Y^{TDR}_k]| = |\mathbb{E}[\delta Y^{DQ}_k]|. \tag{48}$$

Thus identity (28) holds.

**Case 4**: If the target critic values and the true value $Q^\pi$ has the following relationship

$$Q^\pi < \mathbb{E}[Q_{\theta'_1}] < \mathbb{E}[Q_{\theta'_2}], \tag{49}$$

where $\mathbb{E}[Q_{\theta'_2}]$ is expected more overestimated ie $|\mathbb{E}[\Psi^k_{\theta'_1}]| < |\mathbb{E}[\Psi^k_{\theta'_2}]|$ that implies

$$|\mu'_1| < |\mu'_2|. \tag{50}$$

Based on Equation (24) and (7), same with vanilla double $Q$ network, our Twin TD-regularized Critic will pick the target value using $Q_{\theta'_1}$ which both mitigates the larger overestimation bias as:

$$\mathbb{E}[y^{TDR}_k] = \mathbb{E}[y^{DQ}_k] = \mathbb{E}[r_k] + \gamma\mathbb{E}[Q_{\theta'_1}(s_{k+1}, \pi_{\phi'}(s_{k+1}))], \tag{51}$$

which based on Equation (26), the two estimation errors of the target value are

$$|\mathbb{E}[\delta Y^{TDR}_k]| = |\mathbb{E}[\delta Y^{DQ}_k]| = |\mathbb{E}[Q^\pi - y^{TDR}_k]| = |\frac{\gamma}{1-\gamma}\mu'_1|. \tag{52}$$

We have

$$|\mathbb{E}[\delta Y^{TDR}_k]| = |\mathbb{E}[\delta Y^{DQ}_k]|. \tag{53}$$

Thus identity (28) holds. Both methods can mitigate the overestimation error.

Note, the above cases study the relationship of $\mathbb{E}[Q_{\theta'_1}] < \mathbb{E}[Q_{\theta'_2}]$ and by applying same procedure for $\mathbb{E}[Q_{\theta'_1}] > \mathbb{E}[Q_{\theta'_2}]$, $|\mathbb{E}[\delta Y^{TDR}_k]| \leq |\mathbb{E}[\delta Y^{DQ}_k]|$ still valid. Thus Theorem 1 holds.

**Theorem 2.** Let $Q^\pi$ denote the true $Q$ value following the current target policy $\pi$, $Q_{\theta_1}$ be the estimated value. We assume that there exists a step random estimation bias $\psi^k_{\theta_1}$ that is independent of $(s_k, a_k)$ with mean $\mathbb{E}[\psi^k_{\theta_1}] = \mu_1, \mu_1 < \infty$, for all $k$. We assume the policy is updated based on critic $Q_{\theta_1}$ using the deterministic policy gradient (DPG) as in Equation 4. Let $\delta\phi_k$ denote the change in actor parameter $\phi$ updates at stage $k$. Accordingly, we denote this change for TDR as $\delta\phi^{TDR}_k$, vanilla DPG as $\delta\phi^{DPG}_k$, and true change without any approximation error in $Q$ as $\delta\phi^{true}_k$. We then have the following,

$$\begin{cases} \mathbb{E}[\delta\phi^{true}_k] \geq \mathbb{E}[\delta\phi^{TDR}_k] \geq \mathbb{E}[\delta\phi^{DPG}_k] & \text{if } \mathbb{E}[\Psi^k_{\theta_1}] < 0, \\ \mathbb{E}[\delta\phi^{true}_k] \leq \mathbb{E}[\delta\phi^{TDR}_k] \leq \mathbb{E}[\delta\phi^{DPG}_k] & \text{if } \mathbb{E}[\Psi^k_{\theta_1}] \geq 0. \end{cases} \tag{54}$$

**Proof.** With learning rate $\alpha$, the true change of the actor parameters in case without any approximation error in $Q$:

$$\mathbb{E}[\delta\phi^{true}_k] = \alpha\mathbb{E}_{s \sim p_{\pi_{\phi^j}}}\left[\nabla_a Q^\pi(s_k, a_k)|_{a=\pi_{\phi^j}(s)} \nabla_{\phi^j}\pi_{\phi^j}(s)\right]. \tag{55}$$

Consider the estimated critic and the true value follow the relationship in Equation 10. Given the same current policy parameters $\phi^j$, the updated parameters using DPG are:

$$\phi^{j+1}_{DPG} = \phi^j + \alpha \mathbb{E}_{s \sim p_{\pi_{\phi^j}}} \left[ \nabla_a(Q^\pi(s_k, a_k) + \Psi^k_{\theta_1})|_{a=\pi_{\phi^j}(s)} \nabla_{\phi^j} \pi_{\phi^j}(s) \right],$$

$$\mathbb{E}[\delta\phi^{DPG}_k] = \alpha \mathbb{E}_{s \sim p_{\pi_{\phi^j}}} \left[ \nabla_a(Q^\pi(s_k, a_k) + \Psi^k_{\theta_1})|_{a=\pi_{\phi^j}(s)} \nabla_{\phi^j} \pi_{\phi^j}(s) \right]. \tag{56}$$

With an overestimation bias $\mathbb{E}[\Psi^k_{\theta_1}] > 0$, the updates encourage more exploration for the overestimated actions, and with an underestimation bias $\mathbb{E}[\Psi^k_{\theta_1}] < 0$, the updates discourage exploration for the underestimated actions. Both result in suboptimal policies.

However, by using TD-regularized actor, and given the same current policy parameters $\phi^j$, the actor updates with Equation (9) are:

$$\phi^{j+1}_{TDR} = \phi^j + \alpha \mathbb{E}_{s \sim p_{\pi_{\phi^j}}} [\nabla_a(Q^\pi(s_k, a_k) + \Psi^k_{\theta_1} - \rho(\Delta))|_{a=\pi_{\phi^j}(s)} \nabla_{\phi^j} \pi_{\phi^j}(s)],$$

$$\mathbb{E}[\delta\phi^{TDR}_k] = \alpha \mathbb{E}_{s \sim p_{\pi_{\phi^j}}} [\nabla_a(Q^\pi(s_k, a_k) + \Psi^k_{\theta_1} - \rho(\Delta))|_{a=\pi_{\phi^j}(s)} \nabla_{\phi^j} \pi_{\phi^j}(s)]. \tag{57}$$

Similar to Lemma 1, $\mathbb{E}[\Psi^k_{\theta_1}] = \frac{1}{1-\gamma}\mu_1$, and from Equations (8) and (9) we have:

$$\mathbb{E}[\Delta] = \mathbb{E}[Q_{\theta_1^{i+1}}(s_k, a_k)] - \mathbb{E}[(r_k + \gamma Q_{\theta_1^{i+1}}(s_{k+1}, \pi_\phi(s_{k+1})))]$$

$$= \mu_1 \tag{58}$$

by selecting $\rho \leq \frac{1}{1-\gamma}$, we have the following:

$$\begin{cases} 0 \geq \mathbb{E}[\Psi^k_{\theta_1} - \rho\Delta] > \mathbb{E}[\Psi^k_{\theta_1}] & \text{if } \mathbb{E}[\Psi^k_{\theta_1}] < 0, \\ 0 \leq \mathbb{E}[\Psi^k_{\theta_1} - \rho\Delta] \leq \mathbb{E}[\Psi^k_{\theta_1}] & \text{if } \mathbb{E}[\Psi^k_{\theta_1}] \geq 0. \end{cases} \tag{59}$$

Therefore by inspecting Equations (55), (56) and (57), we have:

$$\begin{cases} \mathbb{E}[\delta\phi^{true}_k] \geq \mathbb{E}[\delta\phi^{TDR}_k] \geq \mathbb{E}[\delta\phi^{DPG}_k] & \text{if } \mathbb{E}[\Psi^k_{\theta_1}] < 0, \\ \mathbb{E}[\delta\phi^{true}_k] \leq \mathbb{E}[\delta\phi^{TDR}_k] \leq \mathbb{E}[\delta\phi^{DPG}_k] & \text{if } \mathbb{E}[\Psi^k_{\theta_1}] \geq 0. \end{cases} \tag{60}$$

Thus Theorem 2 holds.

**Theorem 3**. Suboptimal actor updates negatively affect the critic. Specifically, consider actor updates as in Theorem 2, in the overestimation case, we have:

$$\mathbb{E}[Q_{\theta_1}(s_k, \pi_{DPG}(s_k)] \geq \mathbb{E}[Q_{\theta_1}(s_k, \pi_{TDR}(s_k))] \geq \mathbb{E}[Q^\pi(s_k, \pi_{True}(s_k))], \tag{61}$$

and in the underestimation case,

$$\mathbb{E}[Q_{\theta_1}(s_k, \pi_{DPG}(s_k)] \leq \mathbb{E}[Q_{\theta_1}(s_k, \pi_{TDR}(s_k))] \leq \mathbb{E}[Q^\pi(s_k, \pi_{True}(s_k))]. \tag{62}$$

**Proof** Following the analysis of the TD3 (Fujimoto et al., 2018), consider Equation (12) in Theorem 2, we have

$$\begin{cases} \mathbb{E}[\delta\phi^{true}_k] \geq \mathbb{E}[\delta\phi^{TDR}_k] \geq \mathbb{E}[\delta\phi^{DPG}_k] & \text{if } \mathbb{E}[\Psi^k_{\theta_1}] < 0 \text{ Underestimate}, \\ \mathbb{E}[\delta\phi^{true}_k] \leq \mathbb{E}[\delta\phi^{TDR}_k] \leq \mathbb{E}[\delta\phi^{DPG}_k] & \text{if } \mathbb{E}[\Psi^k_{\theta_1}] \geq 0 \text{ Overestimate}. \end{cases} \tag{63}$$

In the overestimation case, the approximate value using TDR and vanilla DPG must be

$$\mathbb{E}[Q_{\theta_1}(s_k, \pi_{DPG}(s_k)] \geq \mathbb{E}[Q_{\theta_1}(s_k, \pi_{TDR}(s_k))] \geq \mathbb{E}[Q^\pi(s_k, \pi_{true}(s_k))]. \tag{64}$$

Similarly, in the underestimation case, the approximate value using TDR and vanilla DPG must be

$$\mathbb{E}[Q_{\theta_1}(s_k, \pi_{DPG}(s_k)] \leq \mathbb{E}[Q_{\theta_1}(s_k, \pi_{TDR}(s_k))] \leq \mathbb{E}[Q^\pi(s_k, \pi_{True}(s_k))]. \tag{65}$$

Thus Theorem 3 holds.

## C  IMPLEMENTATION DETAILS

We use PyTorch for all implementations. All results were obtained using our internal server consisting of AMD Ryzen Threadripper 3970X Processor, a desktop with Intel Core i7-9700K processor, and two desktops with Intel Core i9-12900K processor.

**Training Procedure**.

An episode is initialized by resetting the environment, and terminated at max step $T = 1000$. A trial is a complete training process that contains a series of consecutive episodes. Each trial is run for a maximum of $1 \times 10^6$ time steps with evaluations at every $2 \times 10^4$ time steps. Each task is reported over 10 trials where the environment and the network were initialized by 10 random seeds, $(0 - 9)$ in this study.

For each training trial, to remove the dependency on the initial parameters of a policy, we use a purely exploratory policy for the first 8000 time steps (start timesteps). Afterwards, we use an off-policy exploration strategy, adding Gaussian noise $\mathcal{N}(0, 0.1)$ to each action.

**Evaluation Procedure**.

Every $1 \times 10^4$ time steps training, we have an evaluation section and each evaluation reports the average reward over 5 evaluation episodes, with no exploration noise and with fixed policy weights. The random seeds for evaluation are different from those in training which each trial, evaluations were performed using seeds $(seeds + 100)$.

**Network Structure and optimizer**.

**TD3**.The actor-critic networks in TD3 are implemented by feedforward neural networks with three layers of weights. Each layer has 256 hidden nodes with rectified linear units (ReLU) for both the actor and critic. The input layer of actor has the same dimension as observation state. The output layer of the actor has the same dimension as action requirement with a tanh unit. Critic receives both state and action as input to THE first layer and the output layer of critic has 1 linear unit to produce $Q$ value. Network parameters are updated using Adam optimizer with a learning rate of $10^{-3}$ for simple control problems. After each time step $k$, the networks are trained with a mini-batch of a 256 transitions $(s, a, r, s')$, $(s, a, r', s')$ in case of LNSS, sampled uniformly from a replay buffer $D$ containing the entire history of the agent.

**D4PG**. Same with the actor-critic networks in D4PG are implemented by feedforward neural networks with three layers of weights. Each layer has 256 hidden nodes with rectified linear units (ReLU) for both the actor and critic. The input layer of actor has the same dimension as observation state. The output layer of the actor has the same dimension as action requirement with a tanh unit. Critic receives both state and action as input to THE first layer and the output layer of critic has a distribution with hyperparameters for the number of atoms $l$, and the bounds on the support $(V_{min}, V_{max})$. Network parameters are updated using Adam optimizer with a learning rate of $10^{-3}$. After each time step $k$, the networks are trained with a mini-batch of 256 transitions $(s, a, r, s')$, $(s, a, r', s')$ in case of LNSS, sampled uniformly from a replay buffer $\mathbb{D}$ containing the entire history of the agent.

**SAC**. The actor-critic networks in SAC are implemented by feedforward neural networks with three layers of weights. Each layer has 256 hidden nodes with rectified linear units (ReLU) for both the actor and critic. The input layer of actor has the same dimension as observation state. The output layer of the actor has the same dimension as action requirement with a tanh unit. Critic receives both state and action as input to the first layer and the output layer of critic has 1 linear unit to produce $Q$ value. Network parameters are updated using Adam optimizer with a learning rate of $10^{-3}$ for simple control problems. After each time step $k$, the networks are trained with a mini-batch of a 256 transitions $(s, a, r, s')$, $(s, a, r', s')$ in case of LNSS, sampled uniformly from a replay buffer $\mathbb{D}$ containing the entire history of the agent.

**Hyperparameters**. To keep comparisons in this work fair, we set all common hyperparameters (network layers, batch size, learning rate, discount factor, number of agents, etc) to be the same for comparison within the same methods and different methods.

For TD3, target policy smoothing is implemented by adding $\epsilon \sim \mathcal{N}(0, 0.2)$ to the actions chosen by the target actor-network, clipped to $(-0.5, 0.5)$, delayed policy updates consist of only updating

the actor and target critic network every $d$ iterations, with $d = 2$. While a larger $d$ would result in a larger benefit with respect to accumulating errors, for fair comparison, the critics are only trained once per time step, and training the actor for too few iterations would cripple learning. Both target networks are updated with $\tau = 0.005$.

The TD3 and TD3+TDR used in this study are based on the paper (Fujimoto et al., 2018) and the code from the authors (https://github.com/sfujim/TD3).

| Hyperparameter TD3 | Value |
|---|---|
| Start timesteps | 8000 steps |
| Evaluation frequency | 20000 steps |
| Max timesteps | 1e6 steps |
| Exploration noise | $\mathcal{N}(0, 0.1)$ |
| Policy noise | $\mathcal{N}(0, 0.2)$ |
| Noise clip | $\pm 0.5$ |
| Policy update frequency | 2 |
| Batch size | 256 |
| Buffer size | 1e6 |
| $\gamma$ | 0.99 |
| $\tau$ | 0.005 |
| Number of parallel actor | 1 |
| LNSS-N | 100 |
| Adam Learning rate | 1e-3 |
| regularization factor | 0.7 |

Table 3: TD3 + TDR hyper parameters used for DMC benckmark tasks

The SAC used in this study is based on paper (Haarnoja et al., 2018) and the code is from GitHub (https://github.com/pranz24/pytorch-soft-actor-critic). and the hyperparameter is from Table 4.

| Hyperparameter SAC | Value |
|---|---|
| Start timesteps | 8000 steps |
| Evaluation frequency | 20000 steps |
| Max timesteps | 1e6 steps |
| Exploration noise | $\mathcal{N}(0, 0.1)$ |
| Policy noise | $\mathcal{N}(0, 0.2)$ |
| Noise clip | $\pm 0.5$ |
| Policy update frequency | 2 |
| Batch size | 256 |
| Buffer size | 1e6 |
| $\gamma$ | 0.99 |
| $\tau$ | 0.005 |
| Temperature parameter $\alpha$ | 0.2 |
| Number of parallel actor | 1 |
| LNSS-N | 100 |
| Adam Learning rate | 1e-3 |

Table 4: SAC hyper parameters used for the DMC benckmark tasks

The D4PG used in this study is based on paper (Barth-Maron et al., 2018) and the code is modified from TD3. The hyperparameter is from Table 5.

All Other algorithms are from the same DRL training platform (Tonic RL) (Pardo, 2020) with the same evaluation as the above algorithms.

**Sparse Reward Setup**. 1) Cheetah Run Sparse: Cheetah needs to run forward as fast as possible. The agent gets a reward only after speed exceeds 2.5 $m/s$, making the reward sparse. $r = 1$. That is, if $v >= 2.5$ else $r = 0$.

| Hyperparameter D4PG | Value |
|---|---|
| Start timesteps | 8000 steps |
| Evaluation frequency | 20000 steps |
| Max timesteps | 1e6 steps |
| Exploration noise | $\mathcal{N}(0, 0.1)$ |
| Noise clip | $\pm 0.5$ |
| Batch size | 256 |
| Buffer size | 1e6 |
| $\gamma$ | 0.99 |
| $\tau$ | 0.005 |
| Number of parallel actor | 1 |
| LNSS-N | 100 |
| Adam Learning rate | 1e-3 |
| $V_{max}$ | 100 |
| $V_{min}$ | 0 |
| $l$ | 51 |
| regularization factor | 0.7 |

Table 5: D4PG + TDR hyper parameters used for the DMC benckmark tasks

# D  TDR ALGORITHMS DETAILS

In this section, we show our TDR-based algorithms. TD3-TDR is shown in Algorithm 1, SAC-TDR is shown in Algorithm 2, and D4PG-TDR is shown in Algorithm 3. We mainly add LNSS reward to the sample collection part. In algorithm update part, we mainly modify the target value selection using Equation 7 for regular DRL and Equation (17) for distributional DRL. Additionally, if applicable, we modify the actor gradient based on Equation 9 for regular DRL and Equation (21) for distributional DRL. All codes will be released to GitHub once the paper get accepted.

---

**Algorithm 1** TD3-TDR

---

**Initialize**:

- Critic networks $Q_{\theta_1}$, $Q_{\theta_2}$ and actor-network $\pi_\phi$ with random parameters, $\theta_1, \theta_2, \phi$
- Target networks $\theta_1' \leftarrow \theta_1$, $\theta_2' \leftarrow \theta_2$, $\phi' \leftarrow \phi$,
- an experience buffer $\mathbb{D}$
- a temporary experience buffer $\mathbb{D}'$ with size $N$
- Total training episode $\mathbb{T}$

1. **For** episode = 1, $\mathbb{T}$ **do**
2.     Reset initialize state $s_0$, $\mathbb{D}'$
3.     **For** k = 0, $T$ **do**
4.         Choose an action $a_k$ based on current state $s_k$ and learned policy from $\mathbb{A}$.
5.         Execute the action $a_k$ and observe a new state $s_{k+1}$ with reward signal $r_k$
6.         Store the transition $(s_k, a_k, r_k, s_{k+1})$ in $\mathbb{D}'$
7.         **if** $k + N - 1 \leq T$ **then**
8.             Get earliest memory $(s_0', a_0', r_0', s_1')$ in the $\mathbb{D}'$
9.             Calculate $r'$ based on Equation (23)
10.            Store the transition $(s_0', a_0', r', s_1')$ in $\mathbb{D}$
11.            Clear original transition $(s_0', a_0', r_0', s_1')$ in the $\mathbb{D}'$
12.         **else**
13.            Repeat step 8 to 11 and Calculate $r'$ based on Equation
14.

$$r_k' = \frac{\gamma - 1}{\gamma^{T-k+1} - 1} \sum_{t=k}^{T} \gamma^{t-k} r_t. \tag{66}$$

15.         **end if**
16.         Sample mini-batch data $(s_t, a_t, r_t', s_{t+1})$ from $\mathbb{D}$
17.         Get next action $a_{t+1} \leftarrow \pi_{\phi'}(s_{t+1})$
18.         Target value $y_t$ based on Equation (7)
19.         Update Critics based on Equation 3
20.         **if** k mod Policy Update frequency **then**
21.            Update $\phi$ by Equation 9
22.            Update target networks:
23.            $\theta_\zeta' \leftarrow \tau\theta_\zeta + (1 - \tau)\theta_\zeta'$
24.            $\phi' \leftarrow \tau\phi + (1 - \tau)\phi'$
25.         **end if**
26.     **end for**
27. **end for**

---

---

**Algorithm 2** SAC-TDR

---

**Initialize**:

- Soft value function $V_\Xi$, target Soft value function $V'_\Xi$, Critic networks $Q_{\theta_1}, Q_{\theta_2}$ and actor-network $\pi_\phi$ with random parameters, $\theta_1, \theta_2, \phi$
- Target networks $\Xi' \leftarrow \Xi$
- an experience buffer $\mathbb{D}$
- a temporary experience buffer $\mathbb{D}'$ with size $N$
- Total training episode $\mathbb{T}$

1. **For** episode = 1, $\mathbb{T}$ **do**
2.     Reset initialize state $s_0$, $\mathbb{D}'$
3.     **For** k = 0, $T$ **do**
4.         Choose an action $a_k$ based on current state $s_k$ and learned policy from $\mathbb{A}$.
5.         Execute the action $a_k$ and observe a new state $s_{k+1}$ with reward signal $r_k$
6.         Store the transition $(s_k, a_k, r_k, s_{k+1})$ in $\mathbb{D}'$
7.         **if** $k + N - 1 \leq T$ **then**
8.             Get earliest memory $(s'_0, a'_0, r'_0, s'_1)$ in the $\mathbb{D}'$
9.             Calculate $r'$ based on Equation (23)
10.            Store the transition $(s'_0, a'_0, r', s'_1)$ in $\mathbb{D}$
11.            Clear original transition $(s'_0, a'_0, r'_0, s'_1)$ in the $\mathbb{D}'$
12.         **else**
13.            Repeat step 8 to 11 and Calculate $r'$ based on Equation
14.

$$r'_k = \frac{\gamma - 1}{\gamma^{T-k+1} - 1} \sum_{t=k}^{T} \gamma^{t-k} r_t. \tag{67}$$

15.         **end if**
16.         Sample mini-batch data $(s_t, a_t, r'_t, s_{t+1})$ from $\mathbb{D}$
17.         Get next action $a_{t+1} \leftarrow \pi_{\phi'}(s_{t+1})$
18.         Target value $y_t$ based on Equation (7)
19.         Update Critics based on Equation 3
20.         Update Soft value function based on original SAC formualtion
21.         Update $\phi$ by original SAC formulation
22.         Update target networks:
23.         $\Xi' \leftarrow \tau\Xi + (1 - \tau)\Xi'$
24.     **end for**
25. **end for**

---

---

**Algorithm 3** D4PG-TDR

---
**Initialize**:

- Critic networks $Z_{\theta_1}$, $Z_{\theta_2}$ and actor-network $\pi_\phi$ with random parameters, $\theta_1, \theta_2, \phi$
- Target networks $\theta_1' \leftarrow \theta_1$, $\theta_2' \leftarrow \theta_2$, $\phi' \leftarrow \phi$,
- an experience buffer $\mathbb{D}$
- a temporary experience buffer $\mathbb{D}'$ with size $N$
- Total training episode $\mathbb{T}$

1. **For** episode = 1, $\mathbb{T}$ **do**
2.     Reset initialize state $s_0$, $\mathbb{D}'$
3.     **For** k = 0, $T$ **do**
4.         Choose an action $a_k$ based on current state $s_k$ and learned policy from $\mathbb{A}$.
5.         Execute the action $a_k$ and observe a new state $s_{k+1}$ with reward signal $r_k$
6.         Store the transition $(s_k, a_k, r_k, s_{k+1})$ in $\mathbb{D}'$
7.         **if** $k + N - 1 \leq T$ **then**
8.             Get earliest memory $(s_0', a_0', r_0', s_1')$ in the $\mathbb{D}'$
9.             Calculate $r'$ based on Equation (23)
10.             Store the transition $(s_0', a_0', r', s_1')$ in $\mathbb{D}$
11.             Clear original transition $(s_0', a_0', r_0', s_1')$ in the $\mathbb{D}'$
12.         **else**
13.             Repeat step 8 to 11 and Calculate $r'$ based on Equation
14.

$$r_k' = \frac{\gamma - 1}{\gamma^{T-k+1} - 1} \sum_{t=k}^{T} \gamma^{t-k} r_t. \tag{68}$$

15.         **end if**
16.         Sample mini-batch data $(s_t, a_t, r_t', s_{t+1})$ from $\mathbb{D}$
17.         Get next action $a_{t+1} \leftarrow \pi_{\phi'}(s_{t+1})$
18.         Target distribution based on Equation (17)
19.         Update Critics based on Equation 18
20.         **if** k mod Policy Update frequency **then**
21.             Update $\phi$ by Equation 21
22.             Update target networks:
23.             $\theta_\zeta' \leftarrow \tau\theta_\zeta + (1 - \tau)\theta_\zeta'$
24.             $\phi' \leftarrow \tau\phi + (1 - \tau)\phi'$
25.         **end if**
26.     **end for**
27. **end for**

---

# E NEWLY OBTAINED DATA DURING REBUTTAL TO FURTHER STRENGTHEN WHY WE SAID THAT TDR IS NOVEL AND OUR RESULTS ARE SOTA

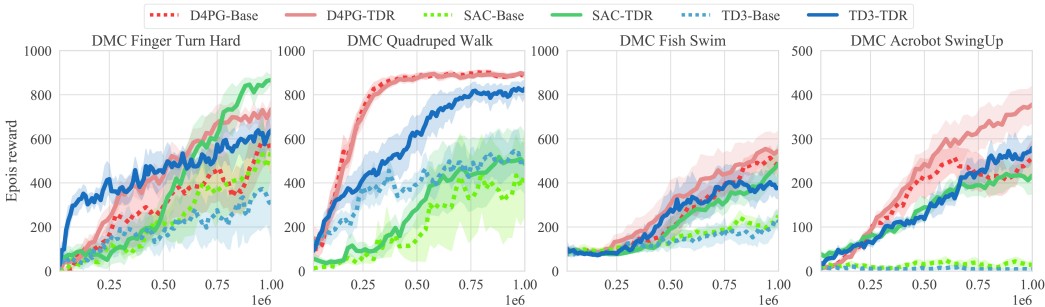

Figure 4: Systematic evaluation of TDR implemented in three SOTA DRL algorithms (SAC, TD3, D4PG) in DMC environments under NOISE FREE environments. The shaded regions represent the 95 % confidence range of respective evaluation over 5 seeds. The x-axis is the number of steps. The results show that our baseline performance is comparable to benchmark results such as those in Pardo (2020) under the SAME hyperparameters. Additionally, TDR continuous to improve performance over baseline methods under NOISE FREE condition.

| | Finger Turn | | Quadruped Walk | | Fish Swim | | Acrobot Swingup | |
|---|---|---|---|---|---|---|---|---|
| | Avg.Rwd | Noise Effect | Avg.Rwd | Noise Effect | Avg.Rwd | Noise Effect | Avg.Rwd | Noise Effect |
| | $[\mu \pm \sigma]$ | [%] | $[\mu \pm \sigma]$ | [%] | $[\mu \pm \sigma]$ | [%] | $[\mu \pm \sigma]$ | [%] |
| SAC-BASE | $515.6 \pm 229.1$ | -87.3 | $419.1 \pm 315.7$ | -53.1 | $229.5 \pm 56.5$ | -68.1 | $15.3 \pm 13.8$ | -73.9 |
| SAC-TDR | $856.5 \pm 59.3$ | -29.7 | $499.4 \pm 258.5$ | -3.9 | $465.4 \pm 136.4$ | -54.4 | $212.6 \pm 63.2$ | -79.8 |
| TD3-BASE | $304.3 \pm 228.3$ | -32.3 | $437.9 \pm 173.7$ | -23.5 | $206.1 \pm 70.5$ | -57.6 | $4.4 \pm 4.1$ | Fail learn |
| TD3-TDR | $612.1 \pm 138.9$ | -6.9 | $824.2 \pm 85.1$ | -42.3 | $372.5 \pm 88.8$ | -45.2 | $274.5 \pm 63.9$ | -81.8 |
| D4PG-BASE | $577.6 \pm 204.4$ | -30.6 | $888.2 \pm 16.1$ | -3.3 | $494.1 \pm 92.8$ | -68.9 | $247.3 \pm 62.9$ | -89.2 |
| D4PG-TDR | $719.7 \pm 70.6$ | -14.4 | $894.1 \pm 18.1$ | -0.7 | $541.5 \pm 155.8$ | -53.8 | $373.6 \pm 70.3$ | -83.2 |

Table 6: Systematic evaluation of TDR implemented in three SOTA DRL algorithms (SAC, TD3, D4PG) in DMC environments under NOISE FREE environments. **Noise Effect** (%) measures performance decay due to introducing significant noise onto observations, actions and rewards (10% each in both directions as in our implementations in this paper). Please note that performance of all baseline methods decay significantly due to added NOISE. However, TDR has helped mitigate such significant performance decay.

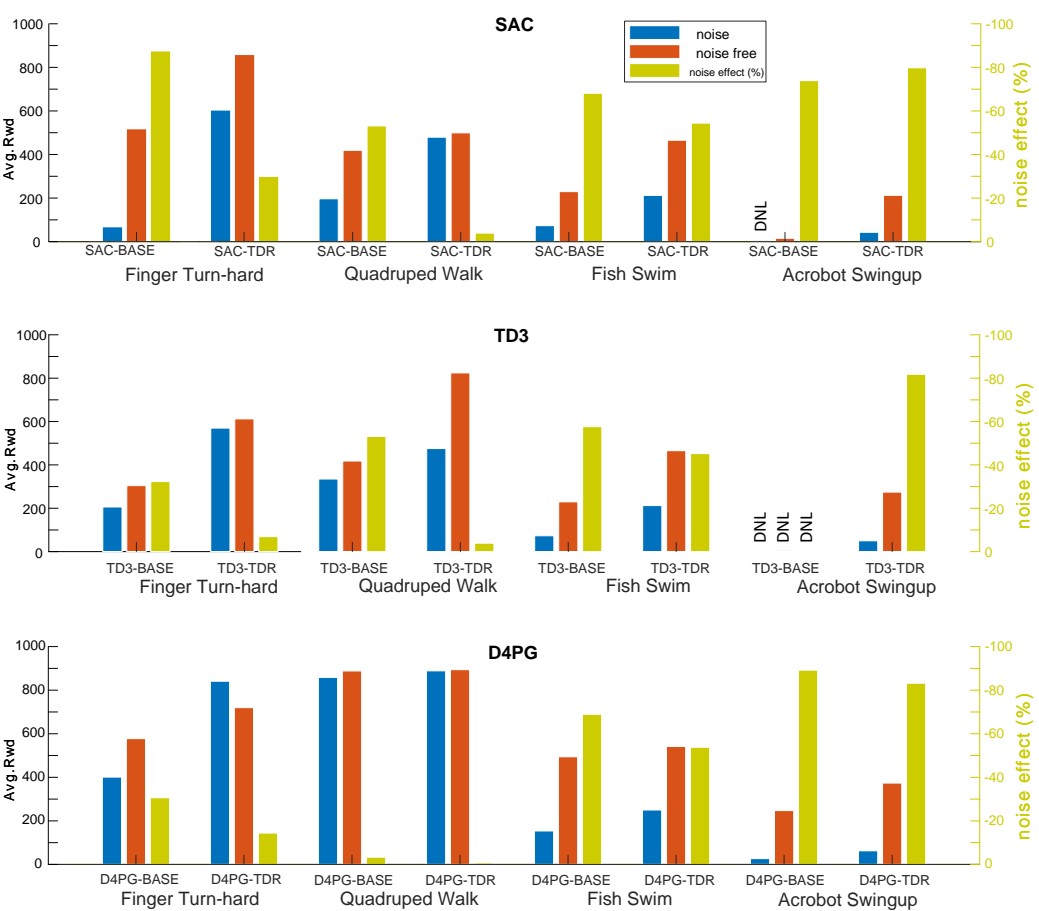

Figure 5: Depiction of some key data in Table 6 and Table 1. Here we illustrate 1) a significant performance drop when introducing the level of noise as we used in the paper to the baseline methods, 2) TDR clearly helps improve upon the performance drop of the baseline methods in the presence of noise. "Noise" refers to significant noise added to observations, actions, and rewards (10% each as in our implementations in this paper). "No noise" refers to implementations in previous works (SAC, TD3, D4PG). "Noise Effect" (%) measures performance decay due to introducing noise.

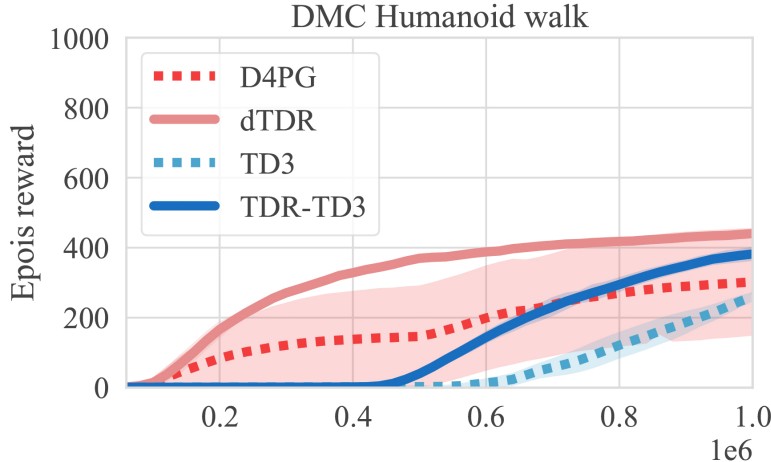

Figure 6: In addition to our extensive evaluations as reported in the paper, here is a new result of TDR on Humanoid walk task (as mentioned by Reviewer SnPL). TDR is implemented in two DRL algorithms (TD3, D4PG) under NOISE FREE condition to be comparable to the literature. The shaded regions represent the 95 % confidence range of evaluations over 3 seeds. The x-axis is the number of steps. This result suggests that when provided with sufficient computational resources, most (good) methods are capable of learning complex humanoid benchmarks. Moreover, please note that TDR continues to enhance the performance of the baseline under these conditions. Lastly, it is important to note that we employed 8 parallel actors in obtaining this result, whereas all other findings presented in Table 6 were obtained from a single actor implementation. In our main paper, We only include results from using single actor implementation since we want to better study baseline algorithms under significant noise and under less abundant computation resources, a common practice that can be found in SOTA publications.

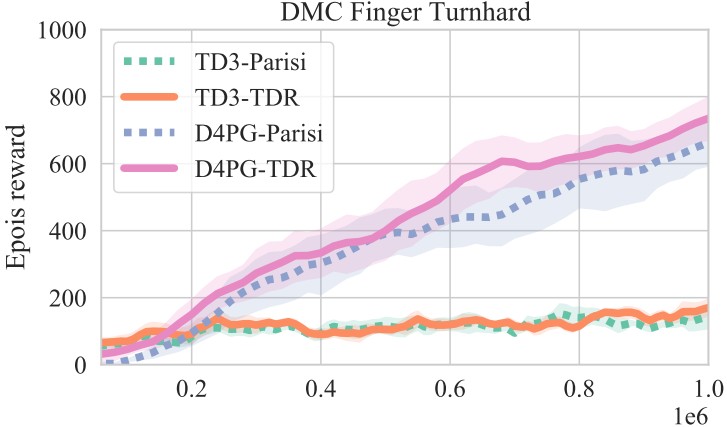

Figure 7: Comparison of Parisi actor and TDR actor. The shaded regions represent the 95 % confidence range of evaluations over 5 seeds. The x-axis is the number of steps. TDR clearly is a novel design, not a simple/trivial extension of Parisi's. Additionally Parisi's study is empirical in nature without any estimation error reduction guarantee. Actually, the paper did not even have a discussion on estimation error. In contrast, in this paper that introduces the new TDR method, our Theorem 2 shows how TD regularized actor can help prevent updates from misleading critics, and Theorem 3 shows how TD regularized actor can mitigate estimation error in critics. Our extensive simulations demonstrate effectiveness of TDR under realistic (and necessary) implementation conditions to study the issue of estimation error.

