# OpenReview forum: "Mitigating Estimation Errors By Twin TD-Regularized Actor and Critic for Deep Reinforcement Learning"
_ICLR.cc/2024/Conference — ICLR 2024 Conference Withdrawn Submission_

### Official Review · Reviewer_SnPL · 2023-10-28

**Soundness:** 1 poor
**Presentation:** 1 poor
**Contribution:** 2 fair
**Rating:** 3
**Confidence:** 4

**Summary:**

The paper introduces the "twin TD-regularized actor- critic (TDR) method" which chooses the target Q-values in temporal difference (TD) learning based on the Q-function attaining the lower TD error, and penalizes the actor network based on the chosen actions' TD errors. This method is combined with priorly proposed practices such as distributional critic functions and LNSS, a slight modified version of n-step returns. The authors evaluate these combined methods on top of  the TD3, SAC, and D4PG algorithms in six environments from the DeepMind Control suite.

**Strengths:**

1) Finding the right balance between pessimism and optimism in TD-based algorithms is a relevant problem that has been the focus of much recent work.

2) The methodological contribution is simple and easy to understand.

3) The experiments are conducted with a fair number of random seeds (10) and include an ablations showing that each of the introduced components contributed positively to the reported performance.

**Weaknesses:**

Major:

1) The experiments are limited to small set of 6 of the easier environments from the DeepMind Control suite. Hence, I believe the empirical results are very far from sufficient for asserting general effectiveness, and appear very much in contrast to the claims made in the paper, e.g., the authors state that one of the main contribution is 'extensive experiments' showing "TDR enables SOTA performance [...] across a wide variety of control tasks" (end of Page 2-Page 3). As a minimum, to make any sort of such claims, I would expect to see results on the full set of DeepMind Control environment, including for the more complex humanoid stand/walk/run tasks.

2) The employed baselines use implementations that have not been optimized for the DeepMind Control suite (nor for the sample-efficiency setting considered as the authors only train for 1000 steps). This makes the reported results notably inferior than what considered standard for what these algorithms can achieve (e.g. see the learning curves in [1]).

3) Section 4  introduces some quite intuitive considerations in the form of convoluted Theorems which I did not find very clear (e.g., in Theorem 1 the authors introduce what they refer to as the "step random estimation bias ψ", but do not use  ψ in any of the main statements of the Theorem. They then re-introduce the same quantity by copying the exact same description in Theorem 2). I would suggest rewriting the Section to convey the paper's consideration more concisely. When using Theorems, I would make sure there is a clear separation between the assumptions and
 the exact theoretical considerations that are being shown (which I found unclear in Theorem 3)

3) The main novelty of the paper is the TDR procedure which foregoes dual TD-learning and chooses the target Q-function based on the lower TD error. However, I believe the paper does not convey a solid intuition as to why this heuristic is superior than standard minimisation of the target Qs. Hence, I am worried the proposed methodology might simply trade-off slightly improved sample-efficiency for stability and convergence (in harder environments/longer training regimes) due to 1) employing n-step returns 2) lowering the pessimism to counteract the actor's Q-maximization. In connection to Point 1 above I would really like to see experimental evidence showing how the proposed algorithms fare in harder environments and for longer training horizons, to empirically validate their efficacy.

4) The authors claim several times that the dual TD-learning (target Q-function minimization from TD3) "promotes a new problem of underestimation, which usually occurs during the early stage of learning, or when subjected to corrupted reward feedback or inaccurate states" (page 4). Yet this claim is not supported by any empirical or theoretical evidence in the text.

Minor:

The authors incorporate  what they refer to as 'Long N-step surrogate state' (LNSS), which appears to be a simple modification to n-step returns that takes the discounted average rather than the discounted sum of the rewards. While they remark several times this is a core component of their methodology, they never explain this simple procedure in the main text and relegate its description to a Section in the Appendix I found unnecessarily convoluted. Am I missing something about this method?

Some typos are still present, I suggest making use of a spell-checker to parse the text (e.g., measureed. page 2)

[1] https://github.com/denisyarats/pytorch_sac

**Questions:**

I would appreciate if the authors could address the questions and criticism I have included in connection to the weaknesses Section above.

---

> ### Author Response · Authors · 2023-11-22
>
> >Benchmark Environment Selection
>
> Please refer to our Common Point 4
>
> >The employed baselines use implementations that have not been optimized for the DeepMind Control suite
>
> The question is fair, BUT we have taken this issue into consideration in our implementations when reporting results. Please read our implementation details of the paper (Appendix C) and Common points 3 more carefully.
> Specifically,
>
> 1. We use default environment setting (1000 steps horizon), which is well accepted as shown in previous papers [1,2,3].
>
> 2. The length of training at 1e6 maximum steps is also well accepted in the community [2,3].
>
> 3. we use the same common hyperparameters for all algorithms. This choice of hyperparameters is well accepted as shown in previous papers [1,2].
>
> As for the reference provided by the reviewer, it uses CUSTOM hyperparameters, while our selected set of hyperparameters are i) consistently used throughout our evaluations, and ii) also used in seminal works such as original TD3 and SAC.
>
>  >Theorem unclear. the authors introduce what they refer to as the "step random estimation bias $\psi$", but do not use $\psi$ in any of the main statements of the Theorem. They then re-introduce the same quantity by copying the exact same description in Theorem 2
>
> In Lemma 1, we show how TD error can be used to measure estimation error by formulating the step estimation bias $\psi$ and the function estimation bias $\Psi$ respectively as
> \begin{equation}
>         \mathbb{E}[\Psi^{k+1} ]  = \sum_{t=k+1}^{\infty} \gamma^{t-k-1} \mathbb{E}[\psi^t_{\theta'_\zeta}].
> \end{equation}
>
> This equation highlights the significance of the step estimation bias $\psi$ in deriving our theorem which can use step bias to  predict function estimation bias in an unbiased way. For a comprehensive understanding of how this bias contributes to our results, we suggest the reviewer read the discussion/proof presented in Common Point 2 and the full proof of Theorems in the Appendix, which includes a further elaboration during rebuttal. This part of our work underscores the integral role of Lemma 1 in supporting the theorems.
>
> Regarding Theorem 3, we build upon the foundation laid in Theorem 2 to make a statement about the impact of suboptimal actor updates on the critic. Specifically, we address how these updates influence the Q value (represented in the critic) in two distinct scenarios: overestimation and underestimation.
> \begin{equation}
>     \mathbb{E}[Q_{\theta_1}(s_k,\pi_{DPG}(s_k)] \geq \mathbb{E}[Q_{\theta_1}(s_k,\pi_{TDR}(s_k))] \geq \mathbb{E}[Q^\pi(s_k,\pi_{True}(s_k))],
> \end{equation}
> and in the underestimation case,
> \begin{equation}
>     \mathbb{E}[Q_{\theta_1}(s_k,\pi_{DPG}(s_k)] \leq \mathbb{E}[Q_{\theta_1}(s_k,\pi_{TDR}(s_k))] \leq \mathbb{E}[Q^\pi(s_k,\pi_{True}(s_k))].
> \end{equation}
> These relationships illustrate how the quality of actor updates, whether suboptimal or not, can significantly affect the critic's performance in terms of Q value estimation. We believe this concept is clearly presented and does not lead to any confusion.
>
> >Novolty and LNSS issue
>
> Please refer to our Common Point 2 for novelty of TDR.
>
> LNSS is not among our claims for novelty in this study. Just like n-step to D4PG, LNSS is a useful add-on.  If the reviewer is interested in LNSS, please refer to their paper [4].
>
> As for the strategy of "lowering the pessimism to counteract the actor's Q-maximization," this approach is central to the functionality of TD3. In TD3, the method involves directly minimizing the Q value, which serves to mitigate the pessimism that arises as a byproduct of the actor's maximization of the Q value. However, as we've identified, this direct minimization approach can lead to other issues, as highlighted by the reviewer. Please refer to our Common Point 2, Theorem 1 (specifically case 1 and case 2) where by using TDR, we effectively addressed this issue by selecting a critic value that is associated with a lesser TD error.
>
> >"promotes a new problem of underestimation, which usually occurs during the early stage of learning, or when subjected to corrupted reward feedback or inaccurate states" (page 4). Yet this claim is not supported by any empirical or theoretical evidence in the text.
>
> Please refer to our Common Point 3. Consider at very early stage of training with $Q = 0$ as a common initialization. If the reward noise is negative, by inspecting the bellman equation, the estimation error will most likely result in an underestimation error. Similarly for inaccurate state and action due to  $s_{noise}$ and $a_{noise}$, it is possible to have $\mathbb{E} [Q(s_{noise},a_{noise})] < \mathbb{E} [Q(s_{true},a_{true}]$ which will result in underestimation. This is why we need to study estimation error under noisy reward, state and action rather than a noise free environment. This also shows the significance and novelty of our work.

---

> ### Author Response · Authors · 2023-11-22
>
> [1] Pardo, F. (2020). Tonic: A deep reinforcement learning library for fast prototyping and benchmarking. arXiv preprint arXiv:2011.07537.
>
> [2] Fujimoto, Scott, Herke Hoof, and David Meger. "Addressing function approximation error in actor-critic methods." International conference on machine learning. PMLR, 2018.
>
> [3] Haarnoja, Tuomas, et al. "Soft actor-critic algorithms and applications." arXiv preprint arXiv:1812.05905 (2018)
>
> [4] Zhong, Junmin, Ruofan Wu, and Jennie Si. "A Long N-step Surrogate Stage Reward for Deep Reinforcement Learning." Thirty-seventh Conference on Neural Information Processing Systems. 2023.

---

### Official Review · Reviewer_KXvd · 2023-10-29

**Soundness:** 3 good
**Presentation:** 2 fair
**Contribution:** 2 fair
**Rating:** 5
**Confidence:** 4

**Summary:**

This paper proposes a new mechanism; instead of directly selecting the minimum value from twin Q values, it selects the target value with smaller TD errors. The authors claim that it could mitigate both overestimation and underestimation issues. The authors implement this mechanism in some popular algorithms and evaluate it in six tasks.

**Strengths:**

I think it is a new try, and this paper gives some theoretical explanations.

**Weaknesses:**

Major Weaknesses lie in the experiments listed in the Questions. Also, the presentation should be improved.

**Questions:**

1. **[Problematic claim in the introduction]** The authors claimed that "The clipped double Q-trick is designed to solve the overestimation caused by the max operator."  Actually, the max operator is not the culprit for the overestimation; we would encounter overestimation even using the bellman evaluation operator. The culprit is function approximation errors.

2. **[Unreliable Baseline Performance]** I think this author's report of SAC and TD3 performances is not consistent with what is reported in other papers.
   For example:

   * AcrobotSwingup task, the author reported SAC performance is ~4@1M steps. However, refer to the TD-MPC[1] paper (Figure 3), ~100@500k steps. **This performance is much higher than the authors' own dTDR algorithm.**

   * Quadruped Walk task, the authors report SAC performance of ~196@1M steps. However, refer to the BAC[2] paper, ~600@400k steps (Figure 32).



3. **[Need More Experimental Results]**  The evaluation of just 6 medium and easy DMControl tasks is not enough to support the effectiveness of this proposed practical mechanism. Add more experimental results on MuJoCo and DMControl tasks.



4. **[Arbitrary Claim on baseline D4PG]** The claim that D4PG is SOTA is an overly arbitrary one in abstract. There are different SOTA methods for different tasks, and I don't agree with the author's claim in the abstract. At least in the BAC [2] I pointed out above, it outperforms D4PG on some tasks. Also, RND[3], and REDQ achieve SOTA on some tasks.



5. **[Add related Works]** BAC is also a paper dedicated to underestimation and overestimation, and I see that it reports good performance and does many experiments on a wide range of benchmark tasks, so I suggest the authors add it to the related work discussion. And possibly do some experimental comparisons if the BAC authors are willing to provide the code or the experimental data.



6. **[Move Table 1 to Appendix ]** Figure 2 and Table 1 are the results of the same set of experiments, and there is no need to occupy half a page in the text.



7. **[Better not to use the limited results to answer Q1-Q6 questions repeatedly]** Your Experiment Section begins by proposing to answer six distinct questions. However, for several of these questions, you only have 1-2 supporting evidence repeatedly from your main experiments. Furthermore, the reader is required to make indirect associations to find these supporting evidence. To enhance the clarity and focus of your work, it may be advisable to pare down the number of questions you aim to address to a more manageable 2-3, then put the unimportant ones in the Appendix and conduct some more experiments to support, not repeatedly use the limited results to answer a lot of questions.

[1] Hansen N, Wang X, Su H. Temporal difference learning for model predictive control[J]. arXiv preprint arXiv:2203.04955, 2022.

[2] Ji T, Luo Y, Sun F, et al. Seizing Serendipity: Exploiting the Value of Past Success in Off-Policy Actor-Critic[J]. arXiv preprint arXiv:2306.02865, 2023.

[3] Burda Y, Edwards H, Storkey A, et al. Exploration by random network distillation[J]. arXiv preprint arXiv:1810.12894, 2018

[4] Chen X, Wang C, Zhou Z, et al. Randomized ensembled double q-learning: Learning fast without a model[J]. arXiv preprint arXiv:2101.05982, 2021.

---

> ### Author Response · Authors · 2023-11-22
>
> >The authors claimed that "The clipped double Q-trick is designed to solve the overestimation caused by the max operator." Actually, the max operator is not the culprit for the overestimation; we would encounter overestimation even using the bellman evaluation operator. The culprit is function approximation errors
>
> We would like to make the following two points.
>
> 1) The notion of overestimation error has long been established and frequently used in  previous seminal works such as those we cited. For example, in Lan 2020 [2], published at ICLR, the first sentence of the second paragraph under section 3 says "The overestimation bias occurs since the target $max_{a' \in A} Q(s_{t+1},a')$ is used in the Q-learning update." TD3 [4] states  in the introduction that "Overestimation bias is a property of Q-learning in which the maximization of a noisy value estimate induces a consistent overestimation". In Thrun \& schwartz 1993 [3], under section 2, it says that "The max operator (in Q learning), however, always picks the largest value, making it particularly sensitive to overestimations."
>
> 2) About the reviewer's comment that "we would encounter overestimation even using the bellman evaluation operator. The culprit is function approximation errors", please refer to our Common Point 3 where we discussed causes/sources of estimation error, which is NOT ONLY just the function approximation error. It can also come from noisy reward, state, and action. This is the very reason of our experiment design, and is considered one of our major contributions to studying estimation error at the root cause.
>
> >Unreliable Baseline Performance
>
> Incorrect assessment. Please refer to our Common Point 3.
>
> >Need More Experimental Results. Add more experimental results on MuJoCo and DMControl tasks.
>
> The reviewer may have missed reading SOTA developments and results on DRL. Please refer to our Common Points 3 and 4. Additionally, please note that Mujoco [1] is indeed a physical simulation engine.  But the DeepMind Control Suite (DMC) has established its position in DRL as providing benchmark environments, which are built upon the Mujoco simulator.
>
> >SOTA Baseline and some related work
>
> First, on using SOTA baseline, please carefully review our Common Point 4.
>
> Additionally, our selection of off-policy baselines, as substantiated by extensive results based on the Deepmind Control Suite [5,6,7], represents the state-of-the-art (SOTA). Specifically, D4PG is widely recognized for its superior performance in continuous control problems, a fact underscored by its impressive scores, frequently surpassing 900 on a scale of 1000. This level of achievement has led to D4PG's extensive use in numerous significant and recent publications as a baseline method for comparison [8,9,10,11].
>
> In regards to a comparison with BAC, please read our Common Point 3. Next, we noted that BAC is currently under review for ICLR 2024 and is yet to be officially published. Our criteria for choosing baselines prioritizes algorithms that are not only published but have also gained widespread acceptance in the research community. This approach ensures that our comparisons are made against benchmarks that are established and validated in the field, and the results can be repeated.
> One more important detail that we'd like to point out - BAC only considers adding a single noise to the action. This does not allow a sufficient study of the estimation error issue.
>
>
> In response to the recommendation to include RND in our comparison, it's important to note that RND primarily focuses on environments such as Montezuma’s Revenge, which is not a continuous control problem. As for REDQ, its evaluation predominantly relies on the OpenAI Gym benchmarks. According to  benchmark studies such as Pardo 2020 [6], it has reached a consensus that  methods such as SAC and PPO, which may perform well in Gym environments, show less impressive results in DMC benchmarks. As such, when consider using the more established DMC benchmarks.  As previously mentioned, D4PG stands out as a well-recognized SOTA method within the community.

---

> ### Author Response · Authors · 2023-11-22
>
> >Move Table 1 to the Appendix
>
> We thank the reviewer for the suggestion. But this table is highly informative, important, and comprehensive as it compliments the learning curves and other plots by providing quantitative performances upon learning convergence.
> This approach is standard and has been a common practice in the community. It can be seen in almost all seminal papers such as  TD3 and SAC.
>
> >Better not to use the limited results to answer Q1-Q6 questions repeatedly
>
> We thank the reviewer for the feedback but we respectfully disagree.
> In quantitative analysis of experimental results employing both horizontal and vertical comparison analyses is necessary and essential. This is a standard technique in scientific data analysis on different performance measures. They answer different questions. Additionally, our presentation of using both horizontal and vertical comparisons actually helps guide readers to read the results and verify our claims.
>
> [1] Mujoco website https://mujoco.org/
>
> [2] Lan, Qingfeng, et al. "Maxmin q-learning: Controlling the estimation bias of q-learning." arXiv preprint arXiv:2002.06487 (2020).
>
> [3] Thrun, S. and Schwartz, A. Issues in using function approximation for reinforcement learning. In Proceedings of the 1993 Connectionist Models Summer School Hillsdale,NJ. Lawrence Erlbaum,1993.
>
> [4] Fujimoto, Scott, Herke Hoof, and David Meger. "Addressing function approximation error in actor-critic methods." International conference on machine learning. PMLR, 2018.
>
> [5] Tassa, Y., Doron, Y., Muldal, A., Erez, T., Li, Y., Casas, D. D. L., ... \& Riedmiller, M. (2018). Deepmind control suite. arXiv preprint arXiv:1801.00690.
>
> [6] Pardo, F. (2020). Tonic: A deep reinforcement learning library for fast prototyping and benchmarking. arXiv preprint arXiv:2011.07537.
>
> [7] Barth-Maron, G., Hoffman, M. W., Budden, D., Dabney, W., Horgan, D., Tb, D., \& Lillicrap, T. (2018). Distributed distributional deterministic policy gradients. arXiv preprint arXiv:1804.08617.
>
> [8] Huang, S., Abdolmaleki, A., Vezzani, G., Brakel, P., Mankowitz, D. J., Neunert, M., \& Hadsell, R. (2022, January). A constrained multi-objective reinforcement learning framework. In Conference on Robot Learning (pp. 883-893). PMLR.
>
> [9] Gulcehre, C., Wang, Z., Novikov, A., Paine, T., Gómez, S., Zolna, K., \& de Freitas, N. (2020). Rl unplugged: A suite of benchmarks for offline reinforcement learning. Advances in Neural Information Processing Systems, 33, 7248-7259.
>
> [10] Agarwal, R., Schwarzer, M., Castro, P. S., Courville, A. C., \& Bellemare, M. (2022). Reincarnating reinforcement learning: Reusing prior computation to accelerate progress. Advances in Neural Information Processing Systems, 35, 28955-28971.
>
> [11] Chen, X., Mu, Y. M., Luo, P., Li, S., \& Chen, J. (2022, June). Flow-based recurrent belief state learning for pomdps. In International Conference on Machine Learning (pp. 3444-3468). PMLR.

---

> ### Comment · Reviewer_KXvd · 2023-11-23
> **Thank the authors for their rebuttal. Here are some kind suggestions.**
>
> Thank you for addressing my comments earlier. I would like to suggest considering the constructive advice offered by the reviewers, as it could enhance the quality of your work. Your responses did not seem to address my recommendations for additional experiments, which I believe are crucial for a more comprehensive evaluation of your approach.
>
> In your paper, you identify a particular method as the state-of-the-art (SOTA) in Reinforcement Learning (RL). However, this claim seems to overlook significant developments in the field, suggesting a potential gap in familiarity with current RL literature. I recommend reviewing the additional references I cited, as they underscore the importance of broad and varied experimentation in algorithmic research.
>
> While I agree that comparing your method with the BAC introduced 5 months ago might not be necessary (it is just a mild suggestion, as shown in my initial tone), I am concerned about the reluctance to benchmark against well-established methods like REDQ or RND. These methods are noted for their superior performance and have been available for over a year, making them relevant comparators.
>
> Furthermore, limiting the scope of experimentation to a few medium and easy tasks does not fully demonstrate the robustness or effectiveness of a proposed method in algorithmic research.
>
> Taking these points into consideration, I have decided to maintain my initial score. I hope these observations will be helpful in guiding the further development of your work.

---

### Official Review · Reviewer_LkhT · 2023-10-30

**Soundness:** 2 fair
**Presentation:** 2 fair
**Contribution:** 2 fair
**Rating:** 3
**Confidence:** 5

**Summary:**

The paper first proposes a TD-Regularized Double Q Networks to effectively control the error of Q-value estimation, and then regularizes the actor based on TD-error to further control the error of Q-value. The paper demonstrates the effectiveness of the method in error control through a large amount of theoretical evidences. The experiments demonstrate that the combination of the method with distributional RL or LNSS also has significant improvement.

**Strengths:**

1) This paper provides a simple yet effective method for mitigating estimation errors. The proposed method is easy to implement.
2) Sufficient theoretical derivations are provided.

**Weaknesses:**

1) The comparative experimental results are not sufficiently reliable. None of the baselines are designed for addressing the estimation error problem.
2) The paper emphasizes that the Twin TD-Regularized method can effectively control errors, but the effectiveness of bias control has not been demonstrated in the experiment section.
3) The ablation experiment is unreasonable, that is, the paper proposes to combine TDR with LNSS method effectively, but LNSS is not the work of this paper and is not suitable for ablation experiments.

[Supplementary review] Thanks for the author's responses to my comments. Based on the comments of all reviewers and the author's reply, I decided to revise my score (Marginally below the acceptance threshold -> Reject, not good enough).

**Questions:**

1) The first Q-network is traditionally chosen to compute the actor TD regularization term. Will it have a negative impact on the selection of Q-network target values in the proposed method?
2) In subsection 3.4, the paper only analyzes the advantages of TDR in theory compared to the previous TD-Regularized actor network method. It should be demonstrated through some experiments.
3) In page 7, the meaning of "TDR has helped successfully address the random initialization challenge caused by random seeds" is confusing and difficult to understand, and it needs further explanation.
4) It is very sensitive to parameters for DRL algorithms, and using different parameters can have different effects. Will the parameters of the comparison algorithms in the experiment be consistent with the original paper (the best situation), and will their performance be worse than the original paper?
5) The paper only demonstrates that TDR can improve the performance of baselines. You should compare some SOTA  bias control algorithms in recent years.
6) The ablation studies are not quite thorough. You should provide more experimental results about approximate estimation error for clearly suggesting the benefits of different components of the algorithm.

---

> ### Author Response · Authors · 2023-11-22
>
> >The comparative experimental results are not sufficiently reliable. None of the baselines are designed for addressing the estimation error problem.
>
> We thank the reviewer for reading our paper and for providing comments.
> However, we respectfully disagree with the reviewer on this point.
> Actually, our experiments were designed for the purpose of addressing estimation error.
> Please refer to our Common Points 2 and 3.
>
> Furthermore, please note that the issue of estimation error is of great importance regardless if the baseline algorithms are designed for this purpose or not. Refer to Common Point 3, estimation error poses a great challenge for almost all DRL algorithms. As demonstrated in our results (Figures 4, 5 and Table 6 in the updated paper pdf during rebuttal), if we deliberately introduce those factors that may cause  estimation error (such as adding noises to $s, a, r$, all baseline algorithms show a significant performance drop from that without noise using the same benchmark environments. However, by using TDR, the lost performance were all  compensated for to a large degree, and the baseline methods with TDR still produce stable results. This clearly shows our contribution and the novelty of TDR.
>
> >Will the parameters of the comparison algorithms in the experiment be consistent with the original paper (the best situation), and will their performance be worse than the original paper
>
> Please refer to our Common Point 3. If we evaluate under the Noise-Free conditions of the original papers, that defeats the purpose of this paper which aims at adequately addressing estimation errors that may come from multiple sources ($s, a, r$, and $\theta$). In real-world applications, these factors are prevalent and ubiquitous.
>
> Nonetheless, in the rebuttal, we have provided new data to further strengthen the novelty and significance of TDR. Please refer to Common Point 1 for our evaluations of baselines without Noise.
>
> >The paper emphasizes that the Twin TD-Regularized method can effectively control errors, but the effectiveness of bias control has not been demonstrated in the experiment section.
>
> We adhere to the same performance measures used in prior works such as TD3 and SAC, both of which aim to address estimation error. Please refer to our Figures 2,4,5, under both noisy and noise free case, TDR boosts average reward of baseline methods.
>
> We also remind the reviewer to read our Common Points 3 where we clearly explained why by adding multiple sources of noise, our baseline results no long replicate those published results (because they did not use noise).
>
> >The first Q-network is traditionally chosen to compute the actor TD regularization term. Will it have a negative impact on the selection of Q-network target values in the proposed method?
>
> Please refer to Common Point 2 where as we discussed that, following the TD3 convention, we use the first critic to compute actor.  This is because theoretically $Q_{\theta_1}$ and $Q_{\theta_2}$  converge to the same result, making it redundant if we go between the two.
>
> For impact on the selection of Q-network value please refer to common point 2.
>
> >In subsection 3.4, the paper only analyzes the advantages of TDR in theory compared to the previous TD-Regularized actor network method. It should be demonstrated through some experiments.
>
> To highlight our points, We are now also providing additional results in the rebuttal Figure 7 of our updated paper. Our TDR actor leads to reduced learning variance, in agreement with the predictions made in Theorem 2. This improvement in our TDR actor's performance may primarily due to the discussion outlined in Common Point 2 that use both online critic can better measure the estimation error of critic updates.
>
> >In page 7, the meaning of "TDR has helped successfully address the random initialization challenge caused by random seeds" is confusing and difficult to understand, and it needs further explanation.
>
> Please refer to one of our cited references by Henderson et al. [1]. In their Figure 5, it is clear that initializing the environment, numpy, and other packages with varying random seeds can lead to results with significant variance in learning performance. Therefore, the ability of a DRL method to effectively overcome this challenge becomes a critical performance evaluation metric.

---

> ### Author Response · Authors · 2023-11-22
>
> >The paper only demonstrates that TDR can improve the performance of baselines. You should compare some SOTA bias control algorithms in recent years.
>
> It's important to note that TD3 and SAC, the two baselines we've compared TDR against, are not only well-accepted in the field but also have demonstrated efficacy in directly addressing overestimation error, a key aspect of bias control. Additionally these two methods have been used as SOTA baselines in most recent top conference papers [4-7].
>
> Our evaluations focus on showing the effectiveness of TDR within the context of general off-policy methods. By applying TDR to these widely recognized and established baseline methods (TD3, SAC, D4PG), we aim to demonstrate its capability to enhance performance in more realistic (aka noisy) environments.
>
> >The ablation studies are not quite thorough. You should provide more experimental results about approximate estimation error for clearly suggesting the benefits of different components of the algorithm. The paper proposes to combine TDR with LNSS method effectively, but LNSS is not the work of this paper and is not suitable for ablation experiments.
>
> We respectfully disagree with the reviewer and here is why.
>
> In our evaluations, we follow the same established protocol in widely accepted and sound research works. We recommend the reviewer  read TD3 and D4PG more carefully to find out. TD3's core innovation lies in its Clipped Double $Q$ Network, accompanied by additional features such as Delayed Policy Update and Target Policy Smoothing Regularization. In their ablation study, the authors of TD3 assessed the impact of each individual component by removing it from the TD3 framework.
>
> Likewise, D4PG's primary advancement is the integration of distributional learning into DDPG, enhanced by additional elements like Prioritized Experience Replay (PER) and n-step updates. The effectiveness of each component in D4PG was evaluated by incrementally adding them to DDPG.
>
> In a similar vein, our approach with TDR involves components like TD regularized critic (TD critic), TD regularized actor (TD actor), and an additional enhancement, namely LNSS. We analyze the contribution of each of these elements by incorporating them into baseline algorithms, thereby assessing their individual impact on performance. This approach allows us to understand the distinct value added by each component in the TDR framework.
>
> In terms of ..."You should provide more experimental results about approximate estimation error...", in addition to our evaluations provided in the paper, please also review our newly added data (Figures 4,5 and Table 6) that summarize TDR evaluations without noise. That actually is an evalution (and performance comparison) for approximation errors.
>
> [1] Henderson, Peter, et al. "Deep reinforcement learning that matters." Proceedings of the AAAI conference on artificial intelligence. Vol. 32. No. 1. 2018.
>
> [2] Fujimoto, Scott, Herke Hoof, and David Meger. "Addressing function approximation error in actor-critic methods." International conference on machine learning. PMLR, 2018.
>
> [3] Haarnoja, Tuomas, et al. "Soft actor-critic algorithms and applications." arXiv preprint arXiv:1812.05905 (2018)
>
> [4] Agarwal, Rishabh, et al. "Reincarnating reinforcement learning: Reusing prior computation to accelerate progress." Advances in Neural Information Processing Systems 35 (2022): 28955-28971.
>
> [5]Eysenbach, Benjamin, et al. "Contrastive learning as goal-conditioned reinforcement learning." Advances in Neural Information Processing Systems 35 (2022): 35603-35620.
>
> [6]Li, Chengshu, et al. "Behavior-1k: A benchmark for embodied ai with 1,000 everyday activities and realistic simulation." Conference on Robot Learning. PMLR, 2023.
>
> [7]Liu, Zuxin, et al. "Constrained variational policy optimization for safe reinforcement learning." International Conference on Machine Learning. PMLR, 2022.

---

> > ### Comment · Reviewer_LkhT · 2023-11-23
> >
> > Thanks for the author's responses to my comments. Based on the comments of all reviewers and the author's reply, I decided to revise my score (Marginally below the acceptance threshold -> Reject, not good enough).

---

### Official Review · Reviewer_2Vrh · 2023-10-31

**Soundness:** 2 fair
**Presentation:** 2 fair
**Contribution:** 1 poor
**Rating:** 3
**Confidence:** 5

**Summary:**

This paper focuses on addressing the problem of estimation bias in DRL. The authors developed the TD-regularized actor-critic (TDR) method, which aims to minimize both overestimation and underestimation errors. The paper also incorporates TDR with other effective DRL techniques, such as distributional learning and the long N-step surrogate stage reward (LNSS) method. The authors evaluate their method with different baselines in the DMC suite.

**Strengths:**

This paper studies how to improve value estimation in DRL, which is a core topic in the RL community. The method is clearly explained.

**Weaknesses:**

The main weaknesses of the paper are the novelty of the approach and the significance of experimental results. It seems a A+B+C work, which modifies previous TD-regularized AC method (Parisi et al., 2019) marginally, and then combines distributional learning and N-step learning into the proposed approach. The experimental part is also a bit weird, where the replicated baselines seem worse than original papers.

**Questions:**

> 3.1 DOUBLE Q IN ACTOR-CRITIC METHOD

This subsection is under the method section (Sec 3). However, the contents in Section 3.1 is actually a summary and description of the literature instead of the proposed method, which can confuse the readers.

> Note from Equation (7) that TDR always uses a target value associated with a smaller target TD value (regardless of the error sign) between the two. ... TDR is naturally positioned to address both overesdiation and underestimation errors.

In fact, the estimation error is measured by comparing the estimated value with an expectation of the true value. How can this sample-based TD error serve as a sound measure for reducing the estimation error? Doesn't this lead to a large variance? In addition, there is a typo in here, where overesdiation should be overestimation.

> Our TD-regularized actor network directly penalizes the actor’s learning objective whenever there is a critic estimation error.

This statement is weird. It would be better to state this as "whenever the critic estimation error is non-zero". In addition, it is unavoidable to have estimation errors in practice when learning these objectives.

---

> ### Author Response · Authors · 2023-11-22
>
> >The main weaknesses of the paper are the novelty of the approach
>
> We thank the reviewer for reading our paper and for providing commons.
> However, we respectfully disagree with the reviewer. The TDR method is NOT an A+B+C work. Please refer to our Common Point 2, especially the one on Novelty of TDR.
>
> As a side note, LNSS is just a useful add-on, playing a role similar to PER in D4PG [3]. We never claimed any credit for LNSS as a (key) contribution of this work.
>
> >Experiment Results Reliable that the replicated baselines seem worse than original papers.
>
> We thank the reviewer for noticing this. Please refer to Common Point 3.
>
> >Section 3.1 is a review and may confuse reader
>
>  Yes, and No. Section 3.1 is not only a  background but also an introduction of necessary notations and problem formulation. This is very common, and necessary, in RL papers. Please refer to TD3, D4PG, SAC [1,2,3].
>
> >How this sample based TD error serve as a sound measure for reducing the estimation error? Doesn't this lead to a large variance?
>
> This sample based approximation is a common approach in DRL. DPG [6] uses sampled  TD error to approximate true gradient. DQN, DDPG, SAC, D4PG [2,3,4,5] use collected samples to approximate $Q$ values. TD3 [1] uses the Q value samples from two target $Q$ networks to address overestimation bias. In a similar vein, our TDR method uses TD error samples from two target networks to approximate both overestimation and underestimation errors.
>
> Regarding the variance, "the variance can grow rapidly with each update if the (estimation) error from each update is not tamed." as state by TD3 authors (refer to TD3 [1] section 5). Based on Theorem 1 and discussions of Common Point 2, we show that by using TDR, RL agents will result in better target values with less estimation error compared to previous clipped double $Q$ network in TD3. Therefore, following the same analysis as TD3, TDR will have the same or even more variance reduction.
>
> >Our TD-regularized actor network directly penalizes the actor’s learning objective whenever there is a critic estimation error.This statement is weird. It would be better to state this as "whenever the critic estimation error is non-zero". In addition, it is unavoidable to have estimation errors in practice when learning these objectives.
>
> We thank the reviewer for rephrasing the sentence for us. Actually there is no difference between how we and the reviewer said it. Nonetheless, we can update the phrase if the paper gets accepted.
>
> [1] Fujimoto, Scott, Herke Hoof, and David Meger. "Addressing function approximation error in actor-critic methods." International conference on machine learning. PMLR, 2018.
>
> [2] Haarnoja, Tuomas, et al. "Soft actor-critic algorithms and applications." arXiv preprint arXiv:1812.05905 (2018)
>
> [3] Barth-Maron, Gabriel, et al. "Distributed distributional deterministic policy gradients." arXiv preprint arXiv:1804.08617 (2018).
>
> [4] Mnih, Volodymyr, et al. "Playing atari with deep reinforcement learning." arXiv preprint arXiv:1312.5602 (2013).
>
> [5] Lillicrap, Timothy P., et al. "Continuous control with deep reinforcement learning." arXiv preprint arXiv:1509.02971 (2015).
>
> [6] Silver, David, et al. "Deterministic policy gradient algorithms." International conference on machine learning. Pmlr, 2014.

---

### Author Response · Authors · 2023-11-22
**Common Points 1 – Our Newly Obtained Data**

> Our new data during rebuttal was obtained on TDR and its baselines under NOISE FREE conditions to highlight why our results (with significant noise) differ from those respective results reported in the literature (noise-free). Specifically, please refer to Appendix E in our updated paper pdf during rebuttal

1. Figure 4 is the evaluation of TDR implemented in three DRL algorithms (SAC, TD3, D4PG) in DMC environments under NOISE FREE environments. The results show that our baseline performance is comparable to benchmark results such as those
in Pardo (2020) under the SAME hyperparameters. Additionally, TDR still improves performance over baseline methods under NOISE FREE conditions.
2. Figure 5 is the visualization of how NOISE can negatively affect baseline performance. We illustrate 1) a significant performance drop when introducing the level of noise as we used in the paper to the baseline methods, 2)  TDR clearly helps improve upon the performance drop of the baseline methods in the presence of noise.
3. Figure 6 is the result of  TDR on a humanoid walk environment under noise free conditions. This indicates that when provided with sufficient computational resources, most methods are capable of learning complex humanoid benchmarks. Moreover, it is noteworthy that TDR continues to enhance the performance of the baseline under these conditions.
4. Figure 7 is a comparison of Parisi's actor regularization and TDR actor regularization. TDR clearly is a novel design, not a simple/trivial extension of Parisi's. Additionally, Parisi's study is empirical in nature without any estimation error reduction guarantee. Actually, the paper did not even have a discussion on estimation error.  In contrast, in this paper that introduces the new TDR method, our Theorem 2 shows how TD regularized actor can help prevent updates from misleading critics, and Theorem 3 shows how TD regularized actor can mitigate estimation error in critics.
5. Table 6 summarizes data of noise free baselines and TDR method. The numerically present performance of all baseline methods decays significantly due to added NOISE. However, TDR has helped prevent such significant performance decay.

---

> ### Author Response · Authors · 2023-11-22
> **Common Points 2 - Novalty of TDR**
>
> > As reviewers recognized that we address a significant estimation bias problem in DRL. Our TDR Is Novel, and Our Results Are SOTA.
>
> > TDR critic:
>
> 1. While one of TD3's key innovations [2] lies in selecting the target value by choosing the minimum of the two target values to reduce overestimation error, just as the TD3 authors state that this method "may lead to an underestimation bias".
>
> 2. Note that this underestimation bias may become particularly detrimental in noisy environments, as observed in our results when a $\pm$ 10\% noise is added to each dimension of the states, actions, and rewards, a case that is more realistic than current SOTA evaluations. **Please refer to the Appendix E in our updated paper for the new Figures 4 and 5, and Table 6.**
>
> 3. While TD3 is an improvement over DQN, our TDR is a further (and significant) improvement over TD3. Notice that this is how TD3 used clipped double $Q$ in its target:
> \begin{equation}
>     y_k = r_k + \gamma \min_{\zeta=1,2} Q_{\theta_\zeta'}(s_{k+1}, \pi_{\phi'}(s_{k+1}))
> \end{equation}
> **The novelty of TDR is in the design of the target value to overcome both over- and under-estimation** (not just overestimation by TD3). For this, we have provided both theoretical and empirical evidence. As TD errors are indicative of estimation errors (as per Lemma 1), we use them from the two target networks to set the target value.  First, the two TD errors from the respective target networks are determined from:
> $$\delta_1 = r_k + \gamma Q_{\theta_1'}(s_{k+1}, \pi_{\phi'}(s_{k+1})) - Q_{\theta_1'}(s_{k}, a_k)$$
> $$\delta_2 = r_k + \gamma Q_{\theta_2'}(s_{k+1}, \pi_{\phi'}(s_{k+1})) - Q_{\theta_2'}(s_{k}, a_k).$$
> The target  value for TDR is then selected from the following:
> if $|\delta_1| \leq |\delta_2|$:
> $$y_k = r_k + \gamma Q_{\theta_1'}(s_{k+1}, \pi_{\phi'}(s_{k+1}))$$
> if $|\delta_1| > |\delta_2|$:
> $$y_k = r_k + \gamma Q_{\theta_2'}(s_{k+1}, \pi_{\phi'}(s_{k+1}))$$
>
> 4) Insights on TDR (analytically). It always uses a target value associated with a smaller target TD value (regardless of the error sign) between the two. Our Lemma 1 shows how TD error can be used to measure the estimation error (Equation (27)). Based on Lemma 1, Case 1 and Case 2 of our Theorem 1 (Equation (35) and Equation (42)) show the benefit of using TDR
> : namely, we can expect less estimation error from TDR than from TD3.
>
> Note, in Theorem 1, we inspect  the relationships between $\mathbb{E}[Q_{\theta_1'}]$ and  $\mathbb{E}[Q_{\theta_2'}]$, through which we discuss all possible scenarios of over- and under-estimations and thus conclude that TDR is an effective means of mitigating both over- and under-estimation.
>
> 5) Insights on TDR (in implementation). Sample based approximation is a common feature in DRL. DPG [1] uses sample TD error to approximate true gradient. DQN, DDPG, SAC, D4PG [3,4,5,6] use collected samples to approximate Q values. In previous work, TD3 [2] uses the Q value samples from two target Q networks to address overestimation bias. In a similar vein, our TDR method uses TD error samples from two target networks.

---

> ### Author Response · Authors · 2023-11-22
> **Common Points 2 - Novalty of TDR**
>
> > TDR actor:
>
> 6) Insights on Parisi's TD regularization method [8].
> The paper is an empirical evaluation of a TD regularized actor aiming at improving learning convergence and expected return. The paper has not discussion on the issue of estimation error. Consequently, there is no analysis and theoretical guarantees on estimation error reduction by using their TD regularized actor.
> As their implementation uses a TD error between the on-line critic and the target critic, this is likely to be the reason they have no results to report in terms of guaranteed estimation error reduction since the two values are from two different networks and they have different estimation error probability distributions.
>
> 7) Significant difference between Parisi's and TDR (theoretically and empirically). TDR actor is a significant advance for off-policy actor-critic methods in DRL.
>
> i) Theoretically, we show how
> TDR helps prevent updates from misleading critics (Theorem 2) and how
> TDR mitigates estimation error in critics (Theorem 3) while Parisi's does not have **ANY**.
>
> ii) ”the variance can grow rapidly with each update if the (estimation) error
> from each update is not tamed.” as state by TD3 authors (refer to TD3 [2] section 5). By address the estimation error, we can get less learning variance and from Figure 7 (empirically), we can clearly see that our TDR actor leads to reduced learning variance, corroborates predictions made in Theorem 2.
> This improvement in our TDR actor’s performance is
> primarily due to the factor outlined in common point 2 that use both online critic can better measure
> the estimation error of critic updates and prevent suboptimal policy update from the misleading critic. (Theorem 2 for detail)
>
> 8) Insights on how the TDR algorithm significantly outperforms Parisi's.
>
> i) TDR critic uses target values to measure TD errors, refer to Equations (5-7) in our paper. No similarity to Parisi's.
>
> ii) In TDR actor, as the value network updates from iteration $i$ to $i+1$, the actor network updates are based on $Q_{\theta^{i+1}}$, NOT $Q_{\theta^{i}}$ as Parisi's. This TD error uses an updated critic, is considered more accurate. Specifically, our TDR actor, tailored for off-policy, uses the following TD error in actor updates:
> \begin{equation}
>     \Delta^{i+1} = Q_{\theta_1^{i+1}}(s_{k}, a_k) - (r_k + \gamma Q_{\theta_1^{i+1}}(s_{k+1}, \pi_{\phi}(s_{k+1}))).
> \end{equation}
> Then the actor can be updated in the direction of maximizing $Q$ while keeping the TD error small,
> \begin{equation}
>     \nabla_\phi J(\phi)
>     = \mathbb{E}[\nabla_{a} (Q_{\theta_1^{i+1}}(s_{k}, a_{k})-  \rho(\Delta^{i+1}))].
> \end{equation}
> In TDR as a convention, same as that in DRL methods based on double Q networks such as TD3,  we only consider update policy using $Q_{\theta_1}$. This is because theoretically $Q_{\theta_1}$ and $Q_{\theta_2}$ eventually converge to the same solution.
>
> 9) In summary, TDR is simple yet effective, and what we have introduced in the paper on TDR is entirely novel.

---

> ### Author Response · Authors · 2023-11-22
> **Common Point 3 - Experiment Design**
>
> 1.  Our primary objective is to address the issue of estimation error. To this very end, we **intentionally** take into account all sources that may contribute to estimation errors. These include noises in observations, actions, and rewards, in addition to neural network approximation error (reflected in network parameter $\theta$) caused estimation error.
> To see that, consider the Bellman equation:
> \begin{equation}
>     Q_\theta(s_k,a_k) = r_k + \gamma Q_\theta(s_{k+1},a_{k+1}),
> \end{equation}
> from which we can see that  four factors can all contribute to estimation biases: 1) reward $r$, 2) state  $s$, 3) action $a$, 4) neural network parameter $\theta$.
>
> > The table below summarizes hyperparameters used in previous work and some other papers suggested by the reviewers. Please note that, **NONE** of these works addresses the issue of estimation error in **ALL FOUR** dimensions ($r$,$s$,$a$, $\theta$).
>
> | Algorithm         | Noise                     | # of actor    | MLP  Hidden Size | Experience Replay |
> |-------------------|---------------------------|---------------|------------------|-------------------|
> | ALL TDR           | $\pm$ 10\% of r,s,a             | 1             | 256              | Uniform           |
> | TD3/SAC  Original | Noise Free                | 1             | 256              | Uniform           |
> | D4PG Original     | Noise Free                | 32            | 256              | PER               |
> | TD-MPC            | Noise Free                | 1             | 512              | PER               |
> | BAC               | Noise Free except white gaussian on action| Not mentioned | 256              | Uniform           |
>
>
> 2. **What is lacking in previous works on addressing the estimation error challenge.**
> Previous works involving TD3, SAC, and D4PG [2,3,4] only focus on addressing estimating error by improving approximation (i.e., doing better estimation by coming up with better parameters $\theta$). Therefore, these results were obtained under a (state, action and reward) noise-free condition.
>
> 3. One of the reviewers mentioned BAC where BAC considers adding a white Gaussian noise to the action only.
> As BAC is a work also under review for ICLR 24, it may be too early to tell how BAC may perform if it is evaluated systematically for all four potential sources  (x, a, r, and $\theta$) contributing to estimation error, a challenge that we are directly addressing now in this paper.
>
> 4) We would like to bring this to reviewers' attention again here: We intentionally wrote in the first sentence under the main evaluation section (Section 5.1) that  "we included a 10\% noise respectively in state, action, and reward in each of the considered DMC environments in order to make the evaluations realistic."
> If we consider each dimension in the state spaces, and that they all have different physical scales (such as velocity, angle in radian, torque, and more), a 10\% noise imposed on each and every physical variable for ALL physical dimensions can be a significant undertaking and excitation to the environments. This makes our realization of the environments highly challenging, and the respective evaluations are thus SOTA.
>
> To be more specific, let $d$  be a uniform noise drawn from
> $U(-0.1,0.1)$. Our  10\% noise (including both directions, i.e., ranging from (-10\%,10\%) is added to each dimension of  $r,a,s$ respectively as follows,
>
> $r_{actual}= r_{true}(1 + d)$
>
> $a_{actual} = a_{true}(1 + d)$
>
> $s_{actual} = s_{true}(1 + d)$
>
> 5) **Indeed, Our Baseline Results Differ from the Published Ones. That Is a Strength, Not a Weakness.**
> The reviewers rightfully said that our baseline results significantly differ from the respective published results. **HOWEVER**, the reviewers **OVERLOOKED** a very important fact that in our experiment design, we intentionally included substantial amount of noise (in x, a, and r) to directly and systematically evaluate the true nature/source of estimation error in order to verify the performance of our TDR method as it aims to address the very issue of estimation error in DRL.
> In another word,
> **this very disparity in  results directly highlights one of the main contributions of our proposed method. Please refer to
> Common Point 1
> for details on the new Figures and Tables for evidence.** In short, please refer to the updated paper (Section E in the Appendices), specifically **Figure 4, Figure 5, and Table 6**, where we can see results of TDR and its corresponding baselines under noise-free conditions.  We observe that now our baseline methods are comparable to the benchmark results [9]. Remarkably, TDR still boosts performance under noise free condition. Additionally, from the "Noise Effect" data in Table 6, adding noise has less effect on TDR than on other base methods. All these point to the great benifit to learning TDR has brought about.

---

> ### Author Response · Authors · 2023-11-22
> **Common Point 4 -- Benchmark Environment Selection**
>
> > Our Selection of Benchmark Environments Is up to Date, Represents a Variety of Tasks, and With Our Intentional 10\% Noise Added, These Benchmarks Are SOTA.
>
> 1) By adding $\pm$ 10\% noise as discussed in Common Point 3, as Figure 5 shows, base methods struggle to maintain their performance as reported (obtained without noise), and fail to learn acrobot swingup. These discrepancies show again that
> **the benchmarks we selected are appropriate and SOTA due to the significant level of noise added**
>  to each and every dimension of $x, r, a$. This is a point we made originally but may have been overlooked.
>
> 2) **Even in the case without our added noise, our selected benchmark environments still are up-to-date, and can be viewed as broadly selected based on the types of environments we used.**
> Among all DMC benchmark environments, the four types of environments are: **"classical control dense", "classical control sparse", "robot locomotion dense", and "robot locomotion sparse"**. For "classical control" types, many recent papers choose acrobot-swingup as discussed in [10].  Additionally, fish-swim [10,11] and finger-turn hard [10,11,12] are also commonly selected "locomotion" benchmark environments. For "classic control sparse" type, we use carpole swingup sparse [13,14], a commonly selected sparse benchmark. For "locomotion sparse" type, as other papers [15,16], we used cheetah to sparsify the dense reward, a common practice in literature [15,16]. We therefore have a "locomotion sparse" benchmark.
>
> 3) Regarding the humanoid benchmark in DMC, many prior works resort to using parallel actors [3,17] for effective training, a method that does not align with our research approach in this study. Consequently, as most recent top conference papers [18,19,11] appeared in 2023 ICML, 2023 AAAI, and 2022 ICML, we have not incorporated this benchmark environment into our analysis.
>
>     However, since one reviewer asked about it, in the rebuttal, we show our TDR is capable of solving humanoid benchmark, we have now included results using 8 parallel actors (**refer to Figure 6 in the Appendix of our updated paper**). For baselines  TD3 and D4PG, TDR still significantly improved performance over base methods.

---

> ### Author Response · Authors · 2023-11-22
> **Common Points Reference**
>
> [1] Silver, David, et al. "Deterministic policy gradient algorithms." International conference on machine learning. Pmlr, 2014.
>
> [2] Fujimoto, Scott, Herke Hoof, and David Meger. "Addressing function approximation error in actor-critic methods." International conference on machine learning. PMLR, 2018.
>
> [3] Barth-Maron, Gabriel, et al. "Distributed distributional deterministic policy gradients." arXiv preprint arXiv:1804.08617 (2018).
>
> [4] Haarnoja, Tuomas, et al. "Soft actor-critic algorithms and applications." arXiv preprint arXiv:1812.05905 (2018).
>
> [5] Mnih, Volodymyr, et al. "Playing atari with deep reinforcement learning." arXiv preprint arXiv:1312.5602 (2013).
>
> [6] Lillicrap, Timothy P., et al. "Continuous control with deep reinforcement learning." arXiv preprint arXiv:1509.02971 (2015).
>
> [7] Van Hasselt, Hado, Arthur Guez, and David Silver. "Deep reinforcement learning with double q-learning." Proceedings of the AAAI conference on artificial intelligence. Vol. 30. No. 1. 2016.
>
> [8] Parisi, Simone, et al. "TD-regularized actor-critic methods." Machine Learning 108 (2019): 1467-1501.
>
> [9] Pardo, Fabio. "Tonic: A deep reinforcement learning library for fast prototyping and benchmarking." arXiv preprint arXiv:2011.07537 (2020).
>
>
> [10] Li, Q., Kumar, A., Kostrikov, I., \& Levine, S. (2023). Efficient Deep Reinforcement Learning Requires Regulating Overfitting. arXiv preprint arXiv:2304.10466.
>
> [11] Cetin, E., \& Celiktutan, O. (2021, July). Learning routines for effective off-policy reinforcement learning. In International Conference on Machine Learning (pp. 1384-1394). PMLR.
>
> [12] Chang, J., Wang, K., Kallus, N., \& Sun, W. (2022, June). Learning bellman complete representations for offline policy evaluation. In International Conference on Machine Learning (pp. 2938-2971). PMLR.
>
> [13] Seyde, T., Gilitschenski, I., Schwarting, W., Stellato, B., Riedmiller, M., Wulfmeier, M., \& Rus, D. (2021). Is bang-bang control all you need? solving continuous control with bernoulli policies. Advances in Neural Information Processing Systems, 34, 27209-27221.
>
> [14] Stooke, A., Lee, K., Abbeel, P., \& Laskin, M. (2021, July). Decoupling representation learning from reinforcement learning. In International Conference on Machine Learning (pp. 9870-9879). PMLR.
>
> [15] Zhang, T., Ren, T., Yang, M., Gonzalez, J., Schuurmans, D., \& Dai, B. (2022, June). Making linear mdps practical via contrastive representation learning. In International Conference on Machine Learning (pp. 26447-26466). PMLR.
>
> [16] Rengarajan, D., Chaudhary, S., Kim, J., Kalathil, D., \& Shakkottai, S. (2022). Enhanced Meta Reinforcement Learning via Demonstrations in Sparse Reward Environments. Advances in Neural Information Processing Systems, 35, 2737-2749.
>
> [17] Tassa, Y., Doron, Y., Muldal, A., Erez, T., Li, Y., Casas, D. D. L., ... \& Riedmiller, M. (2018). Deepmind control suite. arXiv preprint arXiv:1801.00690.
>
>
> [18] Liu, Xuefeng, et al. "Active policy improvement from multiple black-box oracles." International Conference on Machine Learning. PMLR, 2023.
>
> [19] Yue, Yang, et al. "Value-consistent representation learning for data-efficient reinforcement learning." Proceedings of the AAAI Conference on Artificial Intelligence. Vol. 37. No. 9. 2023.

---

### Meta-Review · Area_Chair_HhZR · 2023-12-05

**Metareview:**

This work uses sample-based TD error of the critic as regularization / penalty in off-policy actor critic algorithms. The work claims that the resulting algorithms address the issue of estimation bias in deep RL. Some performance improvements are seen on 6 noisy DeepMind Control environments.

Strengths: The way that this work uses sampled-based TD error of the critic appears novel to my knowledge. The empirical study adds additional noise (randomness) to the original environments, making them more challenging.

Weakness: Sample-based TD error, as well as Bellman error, is known to be a poor proxy for measuring the quality of the value function. This work, however, entirely rests on the use of them a metric and lacks proper discussion regarding its defect. I refer the authors to [1] and  Chapter 11 of [2], as well as references therein for more detailed discussion about this issue. Given the fundamental theoretical defect of the proposed approach, I would like to see much more comprehensive empirical study, including, e.g., all DeepMind Control suite environments with an array of noise levels, before I can be convinced of its empirical effectiveness.

[1] Fujimoto, Scott, et al. "Why should i trust you, bellman? the bellman error is a poor replacement for value error." arXiv preprint arXiv:2201.12417 (2022).
[2] Sutton, Richard S., and Andrew G. Barto. Reinforcement learning: An introduction. MIT press, 2018.

**Justification For Why Not Higher Score:**

This work is based on the idea of using sampled based TD error as a metric for value estimation quality, which is known to be fundamentally flawed, and the authors did not discuss this issue at all.

**Justification For Why Not Lower Score:**

N/A

---

### Decision · Program_Chairs · 2024-01-16

Reject